# Real-time, low-latency closed-loop feedback using markerless posture tracking

**Gary A Kane[1], Gonçalo Lopes[2], Jonny L Saunders[3], Alexander Mathis[1,4], Mackenzie W Mathis[1,4]\***

[1]The Rowland Institute at Harvard, Harvard University, Cambridge, United States; [2]NeuroGEARS Ltd, London, United Kingdom; [3]Institute of Neuroscience, Department of Psychology, University of Oregon, Eugene, United States; [4]Center for Neuroprosthetics, Center for Intelligent Systems, & Brain Mind Institute, School of Life Sciences, Swiss Federal Institute of Technology (EPFL), Lausanne, Switzerland

**Abstract** The ability to control a behavioral task or stimulate neural activity based on animal behavior in real-time is an important tool for experimental neuroscientists. Ideally, such tools are noninvasive, low-latency, and provide interfaces to trigger external hardware based on posture. Recent advances in pose estimation with deep learning allows researchers to train deep neural networks to accurately quantify a wide variety of animal behaviors. Here, we provide a new `DeepLabCut-Live!` package that achieves low-latency real-time pose estimation (within 15 ms, >100 FPS), with an additional forward-prediction module that achieves zero-latency feedback, and a dynamic-cropping mode that allows for higher inference speeds. We also provide three options for using this tool with ease: (1) a stand-alone GUI (called `DLC-Live! GUI`), and integration into (2) `Bonsai`, and (3) `AutoPilot`. Lastly, we benchmarked performance on a wide range of systems so that experimentalists can easily decide what hardware is required for their needs.

**\*For correspondence:**
mackenzie.mathis@epfl.ch

## Introduction

In recent years, advances in deep learning have fueled sophisticated behavioral analysis tools (*Insafutdinov et al., 2016*; *Newell et al., 2016*; *Cao et al., 2017*; *Mathis et al., 2018b*; *Pereira et al., 2019*; *Graving et al., 2019*; *Zuffi et al., 2019*). Specifically, advances in animal pose estimation–the ability to measure the geometric configuration of user-specified keypoints–have ushered in an era of high-throughput quantitative analysis of movements (*Mathis and Mathis, 2020*). One such state-of-the-art animal pose estimation package, DeepLabCut (DLC; *Mathis et al., 2018b*), provides tailored networks that predict the posture of animals of interest based on video frames, and can run swiftly in offline batch processing modes (up to 2500 FPS on standard GPUs; *Mathis et al., 2020a*; *Mathis and Warren, 2018*). This high-throughput analysis has proven to be an invaluable tool to probe the neural mechanisms of behavior (*Mathis and Mathis, 2020*; *von Ziegler et al., 2021*; *Mathis et al., 2020b*). The ability to apply these behavioral analysis tools to provide feedback to animals in real time is crucial for causally testing the behavioral functions of specific neural circuits.

Here, we describe a series of new software tools that can achieve low-latency closed-loop feedback based on animal pose estimation with DeepLabCut (*Figure 1*). Additionally, these tools make it easier for experimentalists to design and conduct experiments with little to no additional programming, to integrate real-time pose estimation with DeepLabCut into custom software applications, and to share previously trained DeepLabCut neural networks with other users. First, we provide a

**Figure 1.** Overview of using DLC networks in real-time experiments within Bonsai, the DLC-Live! GUI, and AutoPilot.

marked speed and latency improvement from existing real-time pose estimation software (*Forys et al., 2020*; *Štih et al., 2019*; *Schweihoff et al., 2019*). Although these tools are also built on top of DeepLabCut, our software optimizes inference code and uses lightweight DeepLabCut models that perform well not only on GPUs, but also on CPUs and affordable embedded systems such as the NVIDIA Jetson platform. Second, we introduce a module to export trained neural network models and load them into other platforms with ease, improving the ability to transfer trained models between machines, to share trained models with other users, and to load trained models in other software packages. Easy loading of trained models into other software packages enabled integration of DeepLabCut into another popular systems neuroscience software, Bonsai (*Lopes et al., 2015*). Third, we provide a new lightweight `DeepLabCut-Live!` package to run DeepLabCut inference online (or offline). This package has minimal software dependencies and can easily be installed on integrated systems, such as the NVIDIA Jetson platform. Furthermore, it is designed to enable easy integration of real-time pose estimation using DeepLabCut into custom software applications. We demonstrate this ability via integration of `DeepLabCut-Live!` into the new AutoPilot framework (*Saunders and Wehr, 2019*).

Using these new software tools, we achieve low latency real-time pose estimation, with delays as low as 10 ms using GPUs and 30 ms using CPUs. Furthermore, we introduce a forward-prediction module that counteracts these delays by predicting the animal's future pose. Using this forward-prediction module, we were able to provide ultra-low latency feedback to an animal (even down to sub-zero ms delay). Such short latencies have only been approachable in marked animals (*Sehara et al., 2019*), but have not been achieved to the best of our knowledge previously with markerless pose

estimation (*Zhao et al., 2019*; *Forys et al., 2020*; *Štih et al., 2019*; *Schweihoff et al., 2019*; *Privitera et al., 2020*).

Lastly, we developed a benchmarking suite to test the performance of these tools on multiple hardware and software platforms. We provide performance metrics for ten different GPUs, two integrated systems and five CPUs across different operation systems. We openly share this benchmarking suite at https://github.com/DeepLabCut/DeepLabCut-live; *Kane, 2020*; copy archived at swh:1:rev:02cd95312ec6673414bdc4ca4c8d9b6c263e7e2f so that users can look up expected inference speeds and run the benchmark on their system. We believe that with more user contributions this will allow the community to comprehensively summarize system performance for different hardware options and can thus guide users in choosing GPUs, integrated systems, and other options for their particular use case.

## Results

### Exporting DeepLabCut models

DeepLabCut enables the creation of tailored neuronal networks for pose estimation of user-defined bodyparts (*Mathis et al., 2018b*; *Nath et al., 2019*). We sought to make these neural networks, which are TensorFlow graphs, easily deployable by developing a model-export functionality. These customized DeepLabCut models can be created from standard trained DeepLabCut models by running the `export_model` function within DeepLabCut (see Materials and methods), or models can be downloaded from the new DeepLabCut Model Zoo.

The model export module utilizes the protocol buffer format (.pb file). Protocol buffers are a language-neutral, platform-neutral extensible mechanism for serializing structured data (https://developers.google.com/protocol-buffers), which makes sharing models simple. Sharing a whole (DeepLabCut) project is not necessary, and an end-user can simply point to the protocol buffer file of a model to run inference on novel videos (online or offline). The flexibility offered by the protocol buffer format allowed us to integrate DeepLabCut into applications written in different languages: a new python package `DeepLabCut-Live!`, which facilitates loading DeepLabCut networks to run inference; and into Bonsai, which is written in C# and runs DeepLabCut inference using TensorFlowSharp (https://github.com/migueldeicaza/TensorFlowSharp).

### A new python package to develop real-time pose estimation applications

The `DeepLabCut-Live!` package provides a simple programming interface to load exported DeepLabCut models and perform pose estimation on single images (i.e., from a camera feed). By design, this package has minimal dependencies and can be easily installed even on integrated systems.

To use the `DeepLabCut-Live!` package to perform pose estimation, experimenters must simply start with a trained DeepLabCut model in the exported protocol buffer format (.pb file) and instantiate a `DLCLive` object. This object can be used to load the DeepLabCut network and perform pose estimation on single images:

```
from dlclive import DLCLive
my_live_object = DLCLive("/exportedmodel/directory")
my_live_object.init_inference(my_image)
pose = my_live_object.get_pose(my_image)
```

On its own, the `DLCLive` class only enables experimenters to perform real-time pose estimation. To use poses estimated by DeepLabCut to provide closed-loop feedback, the `DeepLabCut-Live!` package uses a `Processor` class. A `Processor` class must contain two methods: process and save. The process method takes a pose as an argument, performs some operation, such as giving a command to control external hardware (e.g. to give reward or to turn on a laser for optogenetic stimulation), and returns a *processed* pose. The save method allows the user to save data recorded by the `Processor` in any format the user desires. By imposing few constraints on the `Processor` object, this tool is very flexible; for example, it can be used to read and write from a variety of commonly used data acquisition and input/output devices, including National Instruments devices, Arduino and

Teensy micro-controllers, as well as Raspberry Pis and similar embedded systems. An example `Processor` object that uses a Teensy micro-controller to control a laser for optogenetics is provided in the `DeepLabCut-Live!` package.

We also provide functionality within this package to test inference speed of DeepLabCut networks. This serves to find the bounds for inference speeds an end user can expect given their hardware and pose estimation requirements. Furthermore, there is a method to display the DeepLabCut estimated pose on top of images to visually inspect the accuracy of DeeplabCut networks.

Ultimately, this package is meant to serve as a software development kit (SDK): to be used to easily integrate real-time pose estimation and closed-loop feedback into other software, either that we provide (described below), or integrated into other existing camera capture packages.

## Inference speed using the DeepLabCut-Live! package

Maximizing inference speed is of utmost importance to experiments that require real-time pose estimation. Some known factors that influence inference speed of DeepLabCut networks include (i) the size of the network (*Mathis et al., 2020b*; *Mathis et al., 2020a*), (ii) the size of images (*Mathis and Warren, 2018*), and (iii) the computational power of the hardware (*Mathis and Warren, 2018*).

The `DeepLabCut-Live!` package offers three convenient methods to increase inference speed by reducing the size of images: static image cropping, dynamic cropping around keypoints, and downsizing images (see Materials and methods). These methods are especially important tools, as they enable experimenters to capture a higher resolution, larger 'full frame' view, but increase inference speed by either performing inference only on the portion of the image in which the animal is present (i.e., dynamically crop the image around the animal), or if the entire image is needed, by performing inference on an image with reduced resolution (i.e., a smaller image). To demonstrate the effect of these factors on inference speeds using the `DeepLabCut-Live!` package, we measured inference speeds for two different architectures: DLC-ResNet-50v1 (*Insafutdinov et al., 2016*; *Mathis et al., 2018b*; *Mathis et al., 2020a*) and DLC-MobileNetV2-0.35 (*Mathis et al., 2020a*) and across a range of image sizes using the downsizing method. These tests were performed on a variety of hardware configurations, ranging from NVIDIA GPUs to Intel CPUs on Linux, Windows, and MacOS computers, as well as NVIDIA Jetson developer kits–inexpensive embedded systems with on-board GPUs, and using two different sets of DeepLabCut networks: a dog tracking model with 20 keypoints and a mouse pupil and licking tracking model with eight keypoints.

As expected, inference speeds were faster for the smaller DLC-MobileNetV2-0.35 networks than the larger DLC-ResNet-50v1 networks (*Mathis et al., 2020a*), faster with smaller images, and faster when using more powerful NVIDIA GPUs compared to smaller GPUs or CPUs (*Figure 2*). For example, with the NVIDIA Titan RTX GPU (24 GB) and the NVIDIA GeForce GTX 1080 GPU (8 GB), we achieved inference speeds of $152 \pm 15$ and $134 \pm 9$ frames per second on medium sized images ($459 \times 349$ pixels) using the MobileNetV2-0.35 DeepLabCut network. Full results are presented in *Figure 2*, *Figure 2—figure supplement 1*, and *Table 1*.

### Inference speed, size vs. accuracy

Although reducing the size of images increases inference speed, it may result in reduced accuracy in tracking if the resolution of the downsized image is too small to make accurate predictions, or if the network is not trained to perform inference on smaller images. To test the accuracy of DeepLabCut tracking on downsized images, we trained three DeepLabCut networks on a mouse open-field dataset that has been previously used for DLC accuracy benchmarking ($640 \times 480$ pixels; *Mathis et al., 2018b*). Each network was trained on a different range of image sizes: the default (50–125% of the raw image size), a wider range of image sizes (10–125%), or only on smaller images (10–50%). This was achieved by altering the training scale parameters ('scale_jitter_lo' and 'scale_jitter_hi') in the DeepLabCut training configuration file. We then tested the accuracy of predictions of each network on test images with scale of 12.5–150%, and calculated the root mean square error (RMSE) of predictions compared to human labels (n = 571 images; 50% for train/test) for each test image scale on each network (*Figure 2—figure supplement 2*). Note, that pixel units is the natural metric for reporting accuracy, but in this dataset the width of length of the mouse was $115.7 \pm 0.6$ (mean ± s.e. m., n = 571) to give the reader a sense of scale.

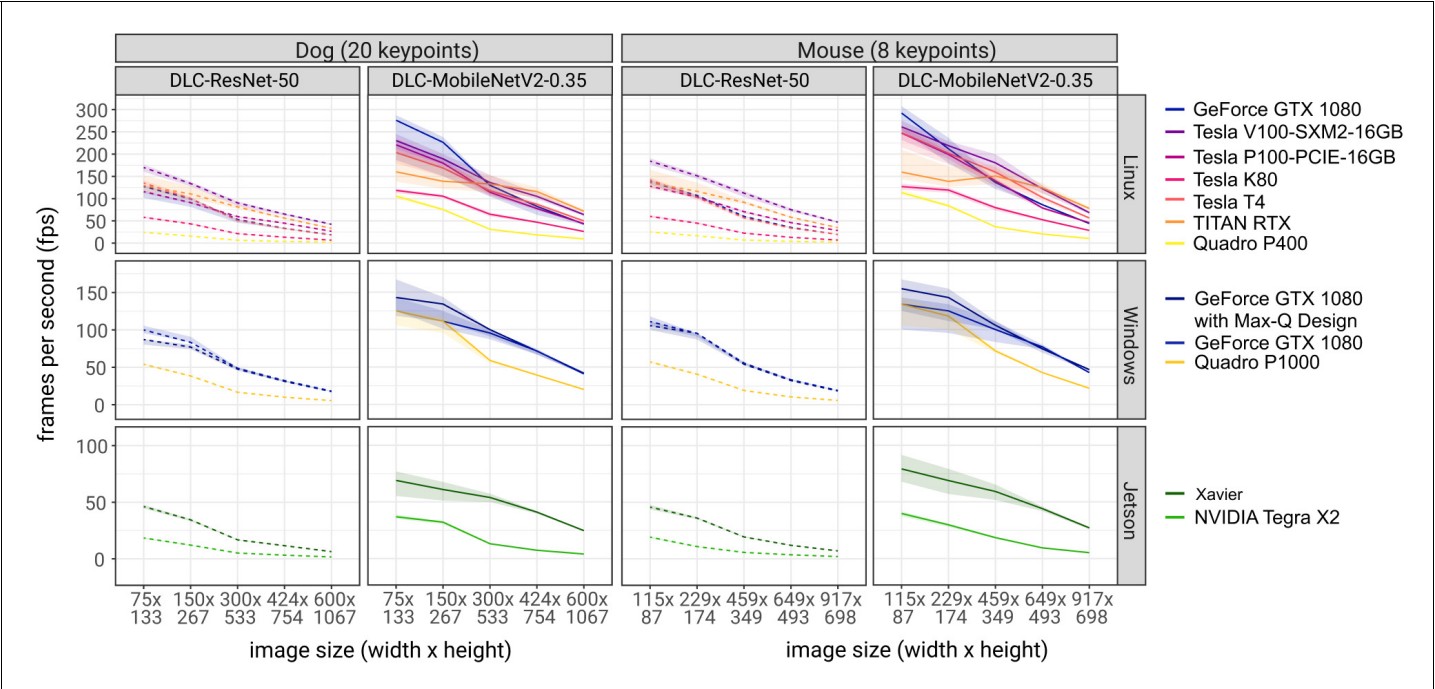

**Figure 2.** Inference speed of different networks with different image sizes on different system configurations. Solid lines are tests using DLC-MobileNetV2-0.35 networks and dashed lines using DLC-ResNet-50 networks. Shaded regions represent 10th-90th percentile of the distributions. N = 1000–10,000 observations per point.

The online version of this article includes the following figure supplement(s) for figure 2:

**Figure supplement 1.** Inference speed of different networks with different image sizes with different system configurations, using CPUs for inference.

**Figure supplement 2.** Accuracy of DLC networks on resized images.

**Figure supplement 3.** Dynamic cropping increased inference speed with little sacrifice to accuracy.

**Figure supplement 4.** The number of keypionts in a DeepLabCut network does not affect inference speed.

Naturally, the best accuracy was achieved on the largest images using networks trained on large images (RMSE = 2.90 pixels at image scale = 150% and training scales of 50–125% or 10–125%). However, there was only minimal loss in accuracy when downsizing images to a scale of 25% of the image size, and these errors were mitigated by using a network trained specifically on smaller images (RMSE = 7.59 pixels at 25% scale with training scale = 10–50%; *Figure 2—figure supplement 2*). However, for the smallest images tested (test scale = 12.5%), accuracy suffered on all networks (RMSE = 26.99–70.92; all RMSE reported as would be in the original image (640 × 480) for comparability), likely because the mouse is only about 10 pixels long. Overall, this highlights that downsizing does not strongly impact the accuracy (assuming the animal remains a reasonable size) and that this is an easy way to increase inference speeds.

## Dynamic cropping: increased speed without effects on accuracy

Another alternative is to not to downsize, but rather to process only the portion of the image that contains the animal. Our approach, which we call 'dynamic cropping', can provide a speed boost (*Figure 2—figure supplement 3*). Thereby the relevant part of the image is selected by taking advantage of predictions from the previous frame, together with the intuition that the animal cannot move arbitrarily fast. Specifically, this method analyzes the portion of the full-size image by calculating a bounding box based on all the body parts from the previous image as well as a margin (note that if the animal is 'lost' in the next frame, the full frame is automatically analyzed). This method captures the animal at it's full resolution (the animal covers the same number of pixels as in the full image), but still runs inference on a smaller image, increasing inference speed.

To examine the effect of dynamic cropping on both inference speed and tracking accuracy, we performed benchmarking on the open source open-field dataset (n = 2330 images *Mathis et al.,*

**Table 1.** Inference speed using the DeepLabCut-live package.

All values are mean ± standard deviation; Frames Per Second (FPS). See also: https://deeplabcut.github.io/DLC-inferencespeed-benchmark/.

| Dog video | Image size (pixels, w*h) | | | | | | |
|---|---|---|---|---|---|---|---|
| | GPU type | DLC model | 75 × 133 | 150 × 267 | 300 × 533 | 424 × 754 | 600 × 1067 |
| Linux | Intel Xeon CPU | MobileNetV2-0.35 | 62 ± 5 | 31 ± 1 | 14 ± 0 | 8 ± 0 | 4 ± 0 |
| | (3.1 GHz) | ResNet-50 | 24 ± 1 | 11 ± 0 | 3 ± 0 | 2 ± 0 | 1 ± 0 |
| | GeForce GTX 1080 | MobileNetV2-0.35 | 256 ± 51 | 208 ± 46 | 124 ± 19 | 80 ± 9 | 43 ± 1 |
| | | ResNet-50 | 121 ± 15 | 95 ± 13 | 52 ± 3 | 33 ± 1 | 19 ± 0 |
| | TITAN RTX | MobileNetV2-0.35 | 168 ± 29 | 144 ± 20 | 133 ± 14 | 115 ± 9 | 71 ± 3 |
| | | ResNet-50 | 132 ± 14 | 113 ± 11 | 82 ± 4 | 56 ± 2 | 33 ± 1 |
| | Tesla V100-SXM2-16GB | MobileNetV2-0.35 | 229 ± 13 | 189 ± 10 | 138 ± 11 | 105 ± 6 | 64 ± 3 |
| | | ResNet-50 | 169 ± 7 | 133 ± 4 | 90 ± 4 | 65 ± 2 | 42 ± 1 |
| | Tesla P100-PCIE-16GB | MobileNetV2-0.35 | 220 ± 12 | 179 ± 9 | 114 ± 7 | 77 ± 3 | 44 ± 1 |
| | | ResNet-50 | 115 ± 3 | 91 ± 2 | 59 ± 2 | 45 ± 1 | 26 ± 1 |
| | Tesla K80 | MobileNetV2-0.35 | 118 ± 4 | 105 ± 3 | 64 ± 4 | 47 ± 2 | 26 ± 1 |
| | | ResNet-50 | 58 ± 2 | 43 ± 1 | 21 ± 1 | 13 ± 0 | 7 ± 0 |
| | Tesla T4 | MobileNetV2-0.35 | 200 ± 17 | 166 ± 13 | 117 ± 10 | 86 ± 5 | 49 ± 2 |
| | | ResNet-50 | 134 ± 8 | 99 ± 5 | 51 ± 3 | 33 ± 1 | 18 ± 0 |
| | Quadro P400 | MobileNetV2-0.35 | 105 ± 4 | 76 ± 2 | 31 ± 1 | 18 ± 0 | 10 ± 0 |
| | | ResNet-50 | 24 ± 0 | 16 ± 0 | 6 ± 0 | 4 ± 0 | 2 ± 0 |
| | Windows | Intel Xeon Silver CPU | | | | | |
| | MobileNetV2-0.35 | 28 ± 1 | 13 ± 0 | 6 ± 0 | 3 ± 0 | 1 ± 0 | |
| | (2.1 GHz) | ResNet-50 | 22 ± 1 | 9 ± 0 | 3 ± 0 | 2 ± 0 | 1 ± 0 |
| | GeForce GTX 1080 | MobileNetV2-0.35 | 142 ± 17 | 132 ± 14 | 98 ± 5 | 69 ± 3 | 41 ± 1 |
| | with Max-Q Design | ResNet-50 | 87 ± 3 | 77 ± 3 | 48 ± 1 | 31 ± 1 | 18 ± 0 |
| | GeForce GTX 1080 | MobileNetV2-0.35 | 128 ± 11 | 115 ± 10 | 94 ± 7 | 72 ± 3 | 41 ± 1 |
| | | ResNet-50 | 101 ± 5 | 86 ± 4 | 49 ± 1 | 32 ± 1 | 18 ± 0 |
| | Quadro P1000 | MobileNetV2-0.35 | 120 ± 11 | 108 ± 10 | 58 ± 2 | 39 ± 1 | 20 ± 0 |
| | | ResNet-50 | 54 ± 2 | 38 ± 1 | 17 ± 0 | 10 ± 0 | 5 ± 0 |
| MacOS | Intel Core i5 CPU | MobileNetV2-0.35 | 39 ± 5 | 20 ± 2 | 11 ± 1 | 7 ± 1 | 4 ± 0 |
| | (2.4 GHz) | ResNet-50 | 8 ± 1 | 4 ± 0 | 1 ± 0 | 1 ± 0 | 0 ± 0 |
| | Intel Core i7 CPU | MobileNetV2-0.35 | 117 ± 8 | 47 ± 3 | 15 ± 1 | 8 ± 0 | 4 ± 0 |
| | (3.5 GHz) | ResNet-50 | 29 ± 2 | 11 ± 1 | 3 ± 0 | 2 ± 0 | 1 ± 0 |
| | Intel Core i9 CPU | MobileNetV2-0.35 | 126 ± 25 | 66 ± 13 | 19 ± 3 | 11 ± 1 | 6 ± 0 |
| | (2.4 GHz) | ResNet-50 | 31 ± 6 | 16 ± 2 | 6 ± 1 | 4 ± 0 | 2 ± 0 |
| Jetson | Xavier | MobileNetV2-0.35 | 68 ± 8 | 60 ± 7 | 54 ± 4 | 41 ± 1 | 25 ± 1 |
| | | ResNet-50 | 46 ± 1 | 34 ± 1 | 17 ± 0 | 12 ± 0 | 6 ± 0 |
| | Tegra X2 | MobileNetV2-0.35 | 37 ± 2 | 32 ± 2 | 13 ± 0 | 7 ± 0 | 4 ± 0 |
| | | ResNet-50 | 18 ± 1 | 12 ± 0 | 5 ± 0 | 3 ± 0 | 2 ± 0 |
| **Mouse Video** | **image size (pixels, w*h)** | | | | | | |
| | GPU Type | DLC Model | 115 × 87 | 229 × 174 | 459 × 349 | 649 × 493 | 917 × 698 |
| Linux | Intel Xeon CPU | MobileNetV2-0.35 | 61 ± 4 | 32 ± 1 | 15 ± 0 | 8 ± 0 | 4 ± 0 |
| | (3.1 GHz) | ResNet-50 | 28 ± 1 | 11 ± 0 | 4 ± 0 | 2 ± 0 | 1 ± 0 |
| | GeForce GTX 1080 | MobileNetV2-0.35 | 285 ± 24 | 209 ± 23 | 134 ± 9 | 86 ± 2 | 44 ± 1 |

*Table 1 continued on next page*

| | | | | | | | |
|---|---|---|---|---|---|---|---|
| | | ResNet-50 | 136 ± 8 | 106 ± 3 | 60 ± 1 | 35 ± 0 | 19 ± 0 |
| | TITAN RTX | MobileNetV2-0.35 | 169 ± 28 | 145 ± 19 | 152 ± 15 | 124 ± 9 | 78 ± 3 |
| | | ResNet-50 | 140 ± 16 | 119 ± 11 | 92 ± 3 | 58 ± 2 | 35 ± 1 |
| | Tesla V100-SXM2-16GB | MobileNetV2-0.35 | 260 ± 12 | 218 ± 9 | 180 ± 18 | 121 ± 8 | 68 ± 3 |
| | | ResNet-50 | 184 ± 6 | 151 ± 5 | 111 ± 6 | 75 ± 3 | 47 ± 2 |
| | Tesla P100-PCIE-16GB | MobileNetV2-0.35 | 246 ± 12 | 198 ± 7 | 138 ± 8 | 79 ± 3 | 46 ± 1 |
| | | ResNet-50 | 128 ± 3 | 103 ± 2 | 70 ± 3 | 46 ± 1 | 28 ± 1 |
| | Tesla K80 | MobileNetV2-0.35 | 127 ± 6 | 119 ± 5 | 79 ± 4 | 52 ± 2 | 28 ± 1 |
| | | ResNet-50 | 60 ± 2 | 45 ± 2 | 23 ± 1 | 13 ± 0 | 7 ± 0 |
| | Tesla T4 | MobileNetV2-0.35 | 242 ± 21 | 197 ± 16 | 156 ± 14 | 101 ± 6 | 56 ± 2 |
| | | ResNet-50 | 141 ± 7 | 102 ± 5 | 57 ± 3 | 34 ± 1 | 20 ± 0 |
| | Quadro P400 | MobileNetV2-0.35 | 114 ± 5 | 84 ± 3 | 37 ± 1 | 20 ± 0 | 10 ± 0 |
| | | ResNet-50 | 25 ± 0 | 16 ± 0 | 7 ± 0 | 4 ± 0 | 2 ± 0 |
| Windows | Intel Xeon Silver CPU | | | | | | |
| | MobileNetV2-0.35 | 28 ± 1 | 14 ± 0 | 6 ± 0 | 3 ± 0 | 1 ± 0 | |
| | (2.1 GHz) | ResNet-50 | 23 ± 1 | 9 ± 0 | 3 ± 0 | 2 ± 0 | 1 ± 0 |
| | GeForce GTX 1080 | MobileNetV2-0.35 | 147 ± 17 | 136 ± 15 | 108 ± 6 | 73 ± 3 | 46 ± 1 |
| | with Max-Q Design | ResNet-50 | 107 ± 5 | 93 ± 4 | 54 ± 2 | 32 ± 1 | 18 ± 0 |
| | GeForce GTX 1080 | MobileNetV2-0.35 | 133 ± 15 | 119 ± 14 | 100 ± 9 | 77 ± 3 | 43 ± 1 |
| | | ResNet-50 | 110 ± 5 | 94 ± 3 | 55 ± 1 | 34 ± 1 | 19 ± 0 |
| | Quadro P1000 | MobileNetV2-0.35 | 129 ± 13 | 115 ± 11 | 72 ± 3 | 43 ± 1 | 22 ± 0 |
| | | ResNet-50 | 57 ± 2 | 41 ± 1 | 19 ± 0 | 10 ± 0 | 6 ± 0 |
| MacOS | Intel Core i5 CPU | MobileNetV2-0.35 | 42 ± 5 | 22 ± 2 | 12 ± 1 | 7 ± 1 | 4 ± 0 |
| | (2.4 GHz) | ResNet-50 | 9 ± 1 | 4 ± 0 | 2 ± 0 | 1 ± 0 | 0 ± 0 |
| | Intel Core i7 CPU | MobileNetV2-0.35 | 127 ± 10 | 48 ± 3 | 16 ± 1 | 9 ± 1 | 4 ± 0 |
| | (3.5 GHz) | ResNet-50 | 30 ± 3 | 10 ± 2 | 3 ± 0 | 2 ± 0 | 1 ± 0 |
| | Intel Core i9 CPU | MobileNetV2-0.35 | 178 ± 15 | 74 ± 16 | 19 ± 3 | 10 ± 1 | 6 ± 0 |
| | (2.4 GHz) | ResNet-50 | 35 ± 8 | 14 ± 2 | 6 ± 1 | 4 ± 0 | 2 ± 0 |
| Jetson | Xavier | MobileNetV2-0.35 | 79 ± 9 | 68 ± 9 | 59 ± 5 | 44 ± 2 | 27 ± 1 |
| | | ResNet-50 | 46 ± 2 | 36 ± 1 | 19 ± 0 | 12 ± 0 | 7 ± 0 |
| | TX2 | MobileNetV2-0.35 | 39 ± 2 | 30 ± 2 | 18 ± 1 | 9 ± 0 | 5 ± 0 |
| | | ResNet-50 | 19 ± 1 | 11 ± 0 | 6 ± 0 | 4 ± 0 | 2 ± 0 |

*2018a*; *Mathis et al., 2018b*). We recorded the DeepLabCut estimated keypoints and inference speed when analyzing the full image (640 × 480 pixels) and when dynamically cropping with a margin of 50, 25, or 10 pixels around the bounding box enclosing all bodyparts from the animal. Dynamic cropping increased inference speed by 75% (full image: 59.6 ± 2.08; dynamic-50: 103.7 ± 19.8; dynamic-25: 110.8 ± 17.6; dynamic-10: 108 ± 25.2; mean ± standard deviation), and resulted in only a small change in tracking performance, with RMSEs of 4.4, 5.5, and 20.6 for dynamic cropping with 50, 25, and 10 pixel margins, respectively (*Figure 2—figure supplement 3*). Qualitatively all predictions apart from the margin 10, looked comparable.

## Number of keypoints does not affect inference speed

Lastly, we tested whether the number of keypoints in the DeepLabCut network affected inference speeds. We modified the dog network to track only the dog's nose (one keypoint), the face (seven keypoints), the upper body (13 keypoints), or the entire body (20 keypoints). Similar to the benchmarking experiments above, we trained a series of networks with both DLC-ResNet-50v1 and DLC-

MobileNetV2-0.35 networks and tested inference speeds on a range of image sizes. The number of keypoints in the model had no effect on inference speed (*Figure 2—figure supplement 4*).

Moreover, we created a website that we aim to continuously update with user input: one can simply export the results of these tests (which capture information about the hardware automatically), and submit the results on GitHub https://github.com/DeepLabCut/DLC-inferencespeed-benchmark. These results, in addition to the extensive testing we provide below, then become a community resource for considerations with regard to GPUs and experimental design https://deeplabcut.github.io/DLC-inferencespeed-benchmark/.

## User-interfaces for real-time feedback

In addition to the DeepLabCut-live package that serves as a SDK for experimenters to write custom real-time pose estimation applications, we provide three methods for conducting experiments that use DeepLabCut to provide closed-loop feedback that do not require users to write any additional code: a standalone user interface called the DLC-Live! GUI (DLG), and by integrating DeepLabCut into popular existing experimental control softwares Autopilot (*Saunders and Wehr, 2019*) and Bonsai (*Lopes et al., 2015*).

## DLC-Live! GUI

The DLG provides a graphical user interface that simultaneously controls capturing data from a camera (many camera types are supported, see Materials and methods), recording videos, and performing pose estimation and closed-loop feedback using the `DeepLabCut-Live!` package. To allow users to both record video and perform pose estimation at the fastest possible rates, these processes run in parallel. Thus, video data can be acquired from a camera without delays imposed by pose estimation, and pose estimation will not be delayed by the time spent acquiring images and saving video data to the hard drive. However, if pose estimation is slower than the frame rate, which will occur if acquiring images at a high frame rate with large images, or if inference is run on less powerful GPUs users can see *Figure 2* and *Table 1* as a guide, image acquisition and pose estimation will not be synchronized. If these processes are not synchronized (i.e. pose estimation is not run as soon as the image is captured), then the delay from image acquisition to obtain the pose consists not only of the time it takes DeepLabCut to perform pose estimation, but also the time from when the image was captured until pose estimation begins. Thus, running image acquisition and pose estimation asynchronously allows users to run pose estimation at the fastest possible rate, but it does not minimize the time from when an image was captured until the pose is measured. If users prefer to minimize the delay from image acquisition to pose estimation, the pose estimation process can wait for the next image to be acquired. In this case, each time a pose is estimated, the pose estimation process will choose to skip a frame to get back in sync with the image acquisition process. Waiting to get back in sync with the pose estimation process will result in a slower rate of pose estimation, and, over the course of an entire experiment, fewer estimated poses. DLG offers users a choice of which mode they prefer: an 'Optimize Rate' mode, in which pose estimation is performed at the maximum possible rate, but there may be delays from the time an image was captured to the time pose estimation begins, and an 'Optimize Latency' mode, in which the pose estimation process waits for a new image to be acquired, minimizing the delay from the time an image was acquired to when the pose becomes available.

To measure the performance of DLG in both modes, we used a video of a head-fixed mouse performing a task that required licking to receive a liquid reward. To simulate a camera feed from an animal in real-time, single frames from the video were loaded (i.e. acquired) at the rate that the video was initially recorded–100 frames per second. We measured three latency periods: (i) the delay from image acquisition to obtaining the pose for each measured pose; (ii) the delay from one measured pose to the next measured pose; and (iii) for each pose in which the tongue was detected, the delay from detecting an action (the mouse's tongue was detected) to turn on an LED (if the tongue was not detected, the LED was not turned on). The presence or absence of the tongue in any image was determined using the likelihood of the tongue keypoint provided by DeepLabCut; if the likelihood was greater than 0.5, the tongue was considered detected. To measure the time from lick detection to turning on an LED, we used a `Processor` object that, when the likelihood of the tongue was greater than 0.5, sent a command to a Teensy micro-controller to turn on an infrared

LED. To determine that the LED had been activated, the status of the LED was read using an infra-red photodetector. When the photodetector was activated, the Teensy reported this back to the `Processor`. The delay from image acquisition to turning on the LED was measured as the difference between the time the frame was acquired and the time that the photodetector had been activated.

This procedure was run under four configurations: pose estimation performed on full-size images (352 × 274 pixels) and images downsized by 50% in both width and height (176 × 137 pixels); both image sizes were run in 'Optimize Rate' and 'Optimize Latency' modes. These four configurations were run on four different computers to span a range of options (to see how they generally perform, please see *Table 1*): a Windows desktop with NVIDIA GeForce 1080 GPU, a Linux desktop with NVI-DIA Quadro P400 GPU, a NVIDIA Jetson Xavier, and a MacBook Pro laptop with Intel Core-i7 CPU.

On a Windows system with GeForce GTX 1080 GPU, DLG achieved delays from frame acquisition to obtaining the pose as fast as 10 ± 1 ms (mean ± sd) in the 'Optimize Latency' mode. Compared to the 'Optimize Latency' mode, this delay was, on average, 4.38 ms longer (95% CI: 4.32–4.44 ms) with smaller images and 4.6 ms longer (95% CI: 4.51–4.63 ms) with larger images in the 'Optimize Rate' mode. As suggested above, the longer delay from frame acquisition to pose in the 'Optimize Rate' mode can be attributed to delays from when images are acquired until pose estimation begins. With a frame acquisition rate of 100 FPS, this delay would be expected to be 5 ms with a range from 0 to 10 ms, as observed.

Running DLG In the 'Optimize Rate' mode on this Windows system, the delay from obtaining one pose to the next was 11 ± 2 ms (rate of 91 ± 11 poses per second) for smaller images and 12 ± 1 ms (rate of 84 ± 9 poses per second) for larger images. Compared to the 'Optimize Rate' mode, the "Optimize Latency" mode was 7.7 ms (95% CI: 7.6–7.8 ms) slower for smaller images and 9.1 ms (95% CI: 9.0–9.1 ms) for larger images. This increased delay from one pose to another can be attributed to time waiting for acquisition of the next image in the 'Optimize Latency' mode.

Lastly, the delay from acquiring an image in which the tongue was detected until the LED could turned on/off includes the time needed to obtain the pose, plus additional time to determine if the tongue is present and to execute the control signal (send a TTL pulse to the LED). To determine the additional delay caused by detection of the tongue and sending a TTL signal to the LED, we compared the delay from image acquisition to turning on the LED with the delay from image acquisition to obtaining a pose in which the LED was not triggered. Detecting the tongue and sending a TTL pulse only took an additional 0.4 ms (95% CI: 0.3–0.6 ms). Thus, the delay from image acquisition to turn on the LED can be almost entirely attributed to pose estimation. Full results from all four tested systems can be found in *Figure 3* and *Table 2*.

## DeepLabCut models in Bonsai

Bonsai is a widely used visual language for reactive programming, real-time behavior tracking, synchronization of multiple data streams and closed-loop experiments (*Lopes et al., 2015*). It is written in C#, thus provides an alternative environment for running real-time DeepLabCut and also test the performance of native TensorFlow inference outside of a Python environment. We developed equivalent performance benchmarks for testing our newly developed Bonsai-DLC plugin https://github.com/bonsai-rx/deeplabcut. This plugin allows loading of the DeepLabCut exported .pb files directly in Bonsai.

We compared the performance of Bonsai-DLC and `DeepLabCut-Live!` on a Windows 10 computer with GeForce GTX 1080 with Max-Q design GPU and found that the performance of running inference through Bonsai-DLC was comparable to `DeepLabCut-Live!` inference (*Figure 4*), suggesting that, as expected, inference speed is limited primarily by available CPU/GPU computational resources rather than by any native language interface optimizations. Moreover, we found the latency to be 34 ms ± 9.5 ms (median, IQR, n = 500) tested at 30 Hz with 384 × 307 pixels, which is equivalent to what was found with DLG above.

We then took advantage of the built-in OpenGL shader support in Bonsai to assess how external load on the GPU would impact DLC inference performance, as would happen when running closed-loop virtual reality simulations in parallel with video inference. To do this, we implemented a simulation of N-body particle interactions using OpenGL compute shaders in Bonsai, where we were able to vary the load on the GPU by changing the number of particles in the simulation, from 5120 up to 51,200 particles. This is a quadratic problem as each particle interacts with every other particle, so it allows us to easily probe the limits of GPU load and its effects on competing processes.

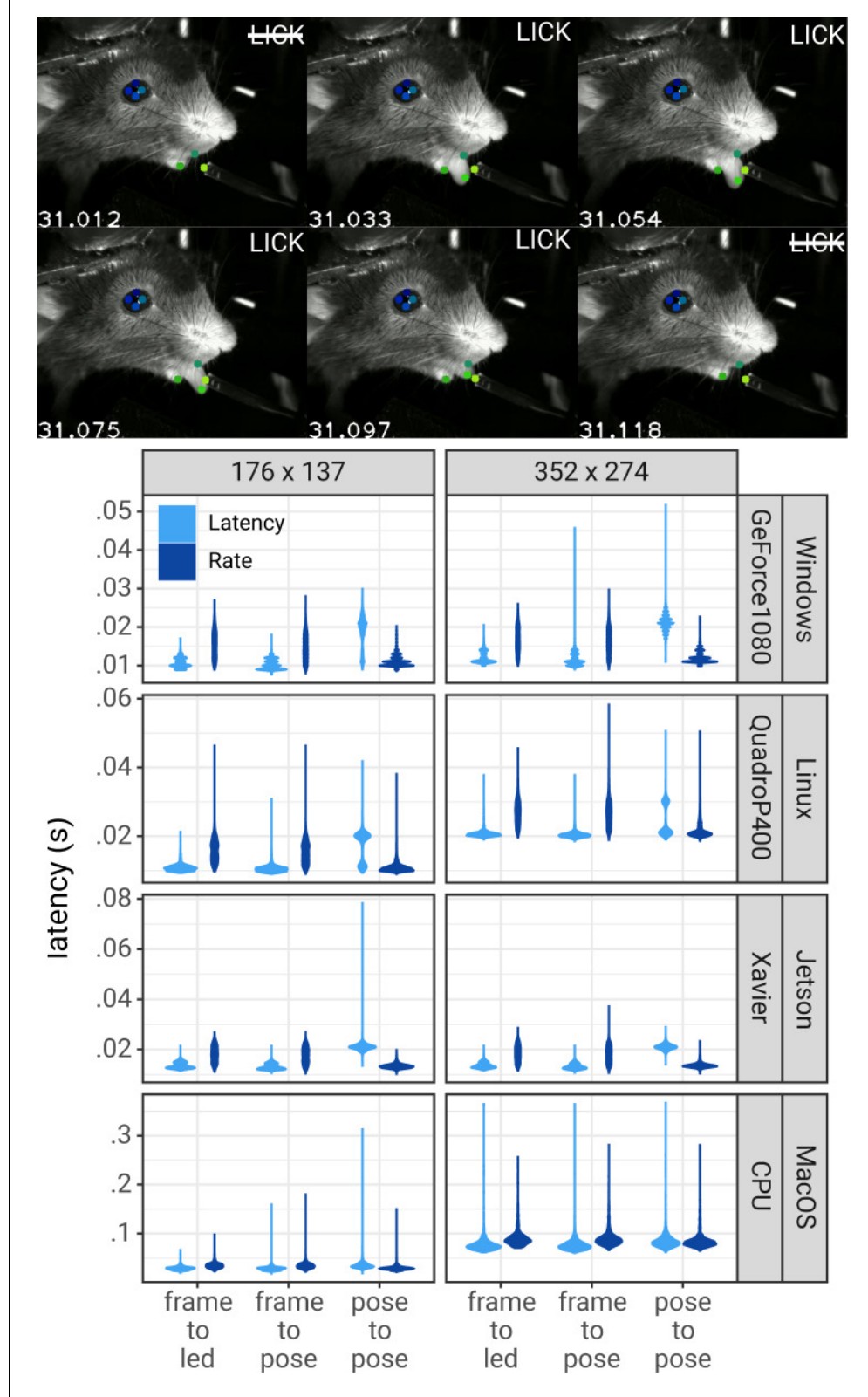

**Figure 3.** Pose estimation latency using the DLC-Live!-GUI. Top: A lick sequence from the test video with eight keypoints measured using the DLC-Live!-GUI (using Windows/GeForce GTX 1080 GPU). Image timestamps are presented in the bottom-left corner of each image. Bottom: Latency from image acquisition to obtaining the pose, from the last pose to the current pose, and from image acquisition when the mouse's tongue was detected to

*Figure 3 continued on next page*

*Figure 3 continued*

turning on an LED. The width of the violin plots indicate the probability density – the likelihood of observing a given value on the y-axis.

Overall, we found that as the number of particle interactions increased the GPU load, there was a corresponding drop in DLC inference speeds (*Figure 4*). The effects of load on inference were non-linear and mostly negligible until the load on the GPU approached 50%, then started to drop as more and more compute banks were scheduled and used, likely due to on-chip memory bottlenecks in the GPU compute units (*Hill et al., 2017*). Nevertheless, as long as GPU load remained balanced, there was no obvious effect on inference speeds, suggesting that in many cases, it would be possible to combine closed-loop accelerated DeepLabCut-live inference with real-time visual environments running on the same GPU (*Lopes et al., 2020*).

## Distributed DeepLabCut with Autopilot

Autopilot is a Python framework designed to overcome problems of simultaneously collecting multiple streams of data by distributing different data streams over a swarm of networked computers (*Saunders and Wehr, 2019*). Its distributed design could be highly advantageous for naturalistic experiments that require large numbers of cameras and GPUs operating in parallel.

Thus, we integrated `DeepLabCut-Live!` into Autopilot in a new module of composable data transformation objects. As a proof of concept, we implemented the minimal distributed case of two computers: one Raspberry Pi capturing images and one NVIDIA Jetson TX2, an affordable embedded system with an onboard GPU, processing them (see Materials and methods, *Table 3*).

We tested the performance of this system by measuring the end-to-end latency of a simple light detection task (*Figure 4A*). The Raspberry Pi lit an LED while capturing and streaming frames to the Jetson. Autopilot's networking modules stream arrays by compressing them on-the-fly with blosc (*Alted et al., 2020*) and routing them through a series of 'nodes'– in this case, each frame passed through four networking nodes in each direction. The Jetson then processed the frames in a chain of `Transform`s that extracted poses from frames using `DLC-Live!` (DLC-MobileNetV2-0.35) and returned a Boolean flag indicating whether the LED was illuminated. `True` triggers were sent back to the Raspberry Pi which emitted a TTL voltage pulse to the LED on receipt.

Experiments were performed with differing acquisition frame rates (30, 60, 90 FPS), and image sizes ($128 \times 128$, $256 \times 256$, $512 \times 416$ pixels; *Figure 5B*). Frame rate had a little effect with smaller images, but at $512 \times 416$, latencies at 30 FPS (median = 161.3 ms, IQR = [145.6–164.4], n = 500) were 38.6 and 52.6 ms longer than at 60 FPS (median = 122.7 ms, IQR = [109.4–159.3], n = 500) and 90 FPS (median = 113.7 ms, IQR = [106.7–118.5], n = 500), respectively.

**Table 2.** Performance of the DeepLabCut-live-GUI (DLG).

F-P = delay from image acquisition to pose estimation; F-L = delay from image acquisition to turning on the LED; FPS (DLG) = Rate of pose estimation (in frames per second) in the DeepLabCut-live-GUI; FPS (DLCLive) = Rate of pose estimation for the same exact configuration directly tested using the DeepLabCut-live benchmarking tool. All values are mean ± STD.

| | | | $176 \times 137$ pixels | | | | $352 \times 274$ pixels | | | |
| | GPU type | Mode | F-P (ms) | F-L (ms) | FPS (DLG) | FPS (DLCLive) | F-P (ms) | F-L (ms) | FPS (DLG) | FPS (DLCLive) |
| --- | --- | --- | --- | --- | --- | --- | --- | --- | --- | --- |
| Windows | GeForce GTX 1080 | Latency | 10 ± 1 | 11 ± 1 | 56 ± 16 | 123 ± 16 | 12 ± 2 | 12 ± 2 | 48 ± 6 | 112 ± 12 |
| | | Rate | 15 ± 3 | 16 ± 3 | 91 ± 11 | | 16 ± 3 | 17 ± 3 | 84 ± 9 | |
| Linux | Quadro P400 | Latency | 11 ± 1 | 11 ± 1 | 63 ± 19 | 105 ± 4 | 20 ± 1 | 21 ± 1 | 42 ± 7 | 52 ± 1 |
| | | Rate | 15 ± 3 | 16 ± 3 | 93 ± 9 | | 27 ± 4 | 27 ± 4 | 47 ± 4 | |
| Jetson | Xavier | Latency | 13 ± 1 | 14 ± 1 | 48 ± 3 | 84 ± 7 | 13 ± 1 | 14 ± 1 | 48 ± 3 | 73 ± 9 |
| | | Rate | 18 ± 3 | 18 ± 3 | 75 ± 5 | | 18 ± 3 | 18 ± 3 | 74 ± 5 | |
| MacOS | CPU | Latency | 29 ± 5 | 29 ± 4 | 29 ± 5 | 62 ± 6 | 79 ± 19 | 79 ± 22 | 12 ± 2 | 21 ± 3 |
| | | Rate | 34 ± 7 | 35 ± 6 | 35 ± 4 | | 91 ± 19 | 92 ± 22 | 12 ± 2 | |

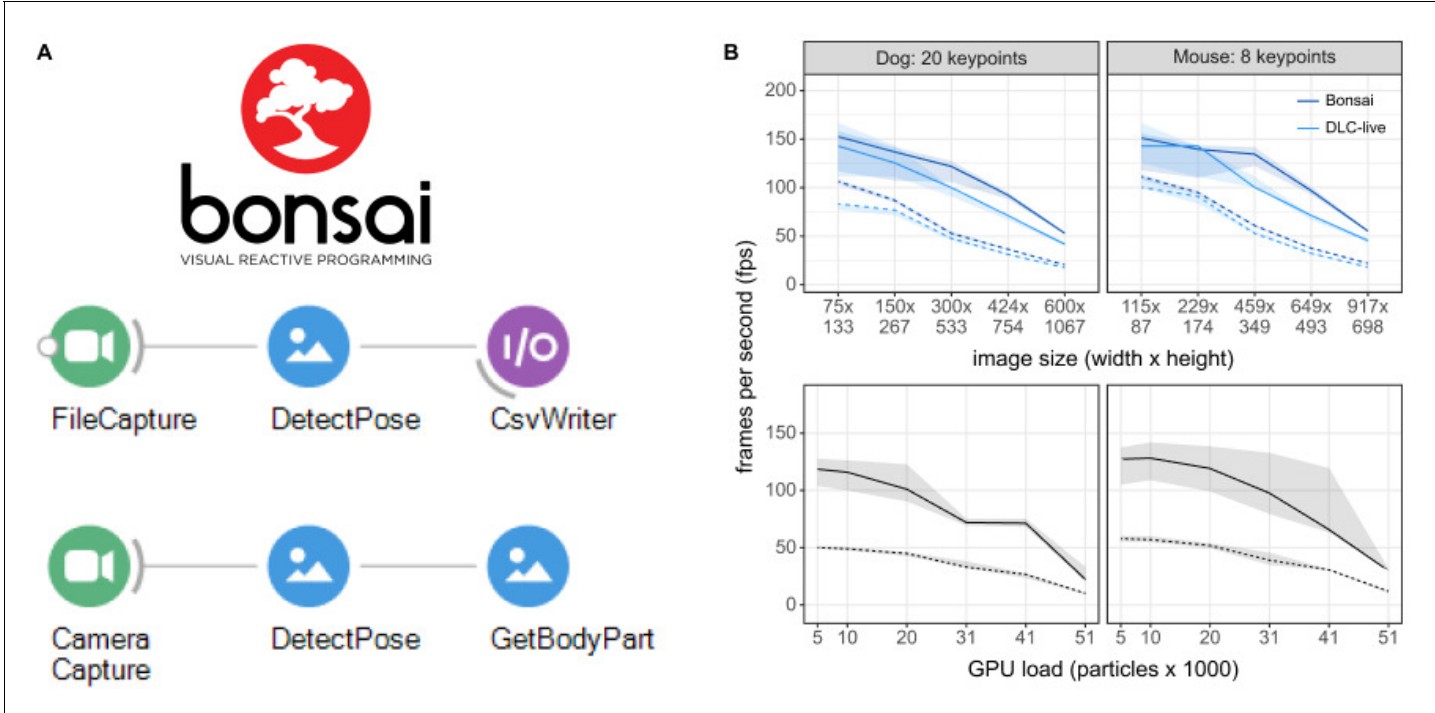

**Figure 4.** Inference speed using the Bonsai-DLC Plugin. Dashed Lines are ResNet-50, solid lines are MobileNetV2-0.35 (**A**) Overview of using DeepLabCut within Bonsai. Both capture-to-write or capture to detect for real-time feedback is possible. (**B**) Top: Direct comparison of the inference speeds using Bonsai-DLC plugin vs. the `DeepLabCut-Live!` package across image sizes from the same computer (OS: Windows 10, GPU: NVIDIA Ge-Force 1080 with Max-Q Design). Bottom: Inference speeds using the Bonsai-DLC plugin while the GPU was engaged in a particle simulation. More particles indicates greater competition for GPU resources.

Frame rate imposes intrinsic latency due to asynchrony between acquired events and the camera's acquisition interval: for example if an event happens at the beginning of a 30 FPS exposure, the frame will not be available to process until 1/30 s = 33.3 ms later. If this asynchrony is distributed uniformly then a latency of half the inter-frame interval is expected for a given frame rate. This quantization of frame rate latency can be seen in the multimodal distributions in *Figure 5B*, with peaks separated by multiples of their inter-frame interval. The inter-frame interval of inference with `Deep-LabCut-Live!` imposes similar intrinsic latency. The combination of these two sources of periodic latency and occasional false-negatives in inference gives a parsimonious, though untested, account of the latency distribution for the 512 × 416 experiments.

Latency at different image sizes were primarily influenced by the relatively slow frame processing of the Jetson TX2 (See *Figure 2*). WIth smaller images (128 × 128 and 256 × 256), inference time (shaded areas in *Figure 5B*) was the source of roughly half of the total latency (inference/median total latency, n = 1500 each, pooled across frame rates. 128 × 128: 32.2/64.6 ms = 49.7%. 256 x 256: 34.5/65.1 ms = 53.0%). At 512 × 416, inference time accounted for between 50% and 70% of

**Table 3.** Materials for Autopilot tests.

| Tool | Version |
| --- | --- |
| Raspberry Pi | 4, 2 GB |
| Autopilot | 0.3.0-2f31e78 |
| Jetson | TX2 Developer Kit |
| Camera | FLIR CM3-U3-13Y3M-CS |
| Spinnaker SDK | 2.0.0.147 |
| Oscilloscope | Tektronix TDS 2004B |

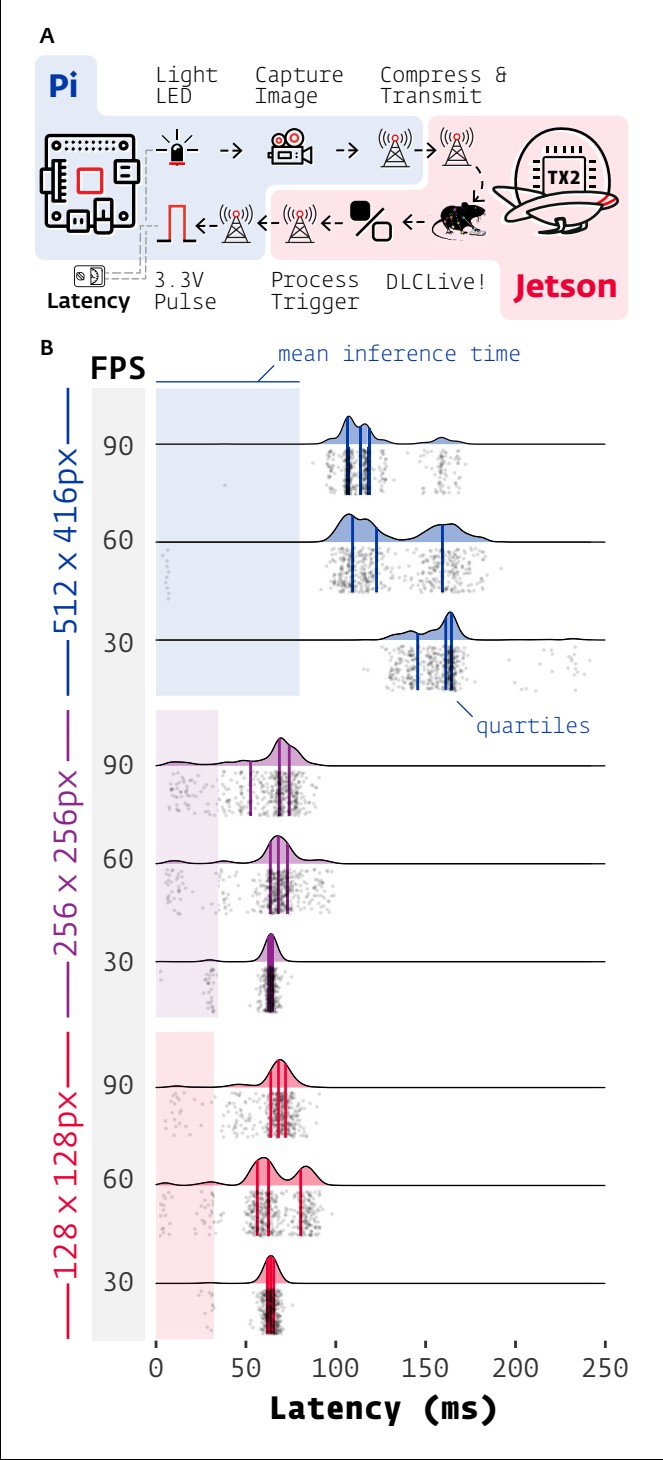

**Figure 5.** Latency of DeepLabCut-Live with Autopilot. (**A**) Images of an LED were captured on a Raspberry Pi 4 (shaded blue) with Autopilot, sent to an Nvidia Jetson TX2 (shaded red) to process with DeepLabCut-Live and Autopilot Transforms, which triggered a TTL pulse from the Pi when the LED was lit. (**B**) Latencies from LED illumination to TTL pulse (software timestamps shown, points and densities) varied by resolution (color groups) and acquisition frame rate (shaded FPS bar). Processing time at each stage of the chain in (**A**) contributes to latency, but pose inference on the relatively slow TX2 (shaded color areas, from *Figure 2*) had the largest effect. Individual measurements (n = 500 each condition) cluster around multiples of the inter-frame interval (eg. 1/30 FPS = 33.3 ms).

total latency (512 × 416, 30 FPS: 79.9/161.3ms = 49.5%, 90 FPS: 79.9/113.7 ms = 70.3%). Minimizing instrument cost for experimental reproducibility and scientific equity is a primary design principle of Autopilot, so while noting that it would be trivial to reduce latency by using a faster GPU or the lower-latency Jetson Xavier (i.e. see above sections), we emphasize that `DeepLabCut-Live!` in Autopilot is very usable with $350 of computing power (TX2 with education discount: $299, Jetson TX2 with NVIDIA Exclusive Education Discount: https://web.archive.org/web/20200723081533/; https://www.nvidia.com/en-us/autonomous-machines/jetson-tx2-education-discount/; Raspberry Pi 4 2 GB RAM: $35, Raspberry Pi from Adafruit: https://web.archive.org/web/20200426114340/; https://www.adafruit.com/product/4292).

Autopilot, of course, imposes its own latency in distributing frame capture and processing across multiple computers. Subtracting latency that is intrinsic to the experiment (frame acquisition asynchrony and GPU speed), Autopilot has approximately 25 ms of overhead (median of latency - mean DeepLabCut inference time - 1/2 inter-frame interval = 25.4 ms, IQR = [14.7–37.9], n = 4500). Autopilot's networking modules take 6.9 ms on average to route a message one-way (*Saunders and Wehr, 2019*) and have been designed for high-throughput rather than low-latency (message batching, on-the-fly compression).

## Real-time feedback based on posture

Lastly, to demonstrate practical usability of triggering a TTL signal based on posture, we performed an experiment using DLG on a Jetson Xavier in which an LED was turned on when a dog performed a 'rearing' movement (raised forelimbs in the air, standing on only hindlimbs; *Figure 6*). First, a DeepLabCut network based on the ResNet-50 architecture (DLC-ResNet-50v1) was trained to track 20 keypoints on the face, body, forelimbs, and hindlimbs of a dog (see Materials and methods). Next, the Jetson Xavier running DLG was used to record the dog as she performed a series of 'rearing' movements in response to verbal commands, with treats given periodically by an experimenter. Video was recorded using a Logitech C270 webcam, with 640 × 480 pixel images at 30 FPS. Inference was run on images downsized by 50% (320 × 240 pixels), using the 'Optimize Rate' mode in DLG.

The dog was considered to be in a 'rearing' posture if the vertical position of at least one of the elbows was above the vertical position of the withers (between the shoulder blades). Similar to the mouse experiment, a `Processor` was used to detect 'rearing' postures and control an LED via communication with a Teensy micro-controller (*Figure 6A*). The LED was turned on upon the first image in which a 'rearing' posture was detected, and subsequently turned off upon the first image in which the dog was not in a 'rearing' posture (for a fully closed-loop stimulus) (*Figure 6B*).

This setup achieved a rate of pose estimation of 22.417 ± 0.928 frames per second, with an image to pose latency of 61 ± 10 ms (N = 1848 frames) and, on images for which the LED was turned on or off, an image to LED latency of 59 ± 11 ms (N = 9 'rearing' movements). However, using DLG, if the rate of pose estimation is slower than video acquisition, not all images will be used for pose estimation (N = 2433 total frames, 1848 poses recorded). To accurately calculate the delay from the ideal time to turn the LED on or off, we must compare the time of the first frame in which a 'rearing' posture was detected from all images recorded, not only from images used for pose estimation. To do so, we estimated the pose on all frames recorded offline using the same exported DeepLabCut model and calculated the ideal times that the LED would have been turned on or off from all available images. According to this analysis, there was a delay of 70 ± 23 ms to turn the LED on or off (consistent with estimates on Jetson systems shown above, see *Figure 3*).

As shown above, there are two methods that could reduce these delays: (i) training a DeepLabCut model based on the MobileNetV2 architecture (DLC-MobileNetV2-0.35 vs DLC-ResNet-50v1) or (ii) using more computationally powerful GPU-accelerated hardware for pose estimation (see *Figure 2*). However, no matter how fast the hardware system, there will be some delay from acquiring images, estimating pose, and providing feedback. To overcome these delays, we developed another method to reduce latency for highly sensitive applications–to perform a forward prediction, or predict the animal's future pose before the next image is acquired and processed. Depending on the forward prediction model, this could potentially reach zero-latency feedback levels (or below)– a dream for experimentalists who aim to study timing of causal manipulations in biological systems.

To reduce the delay to turn on the LED when the dog exhibited a 'rearing' movement, we implemented a Kalman filter that estimated the position, velocity, and acceleration of each keypoint, and

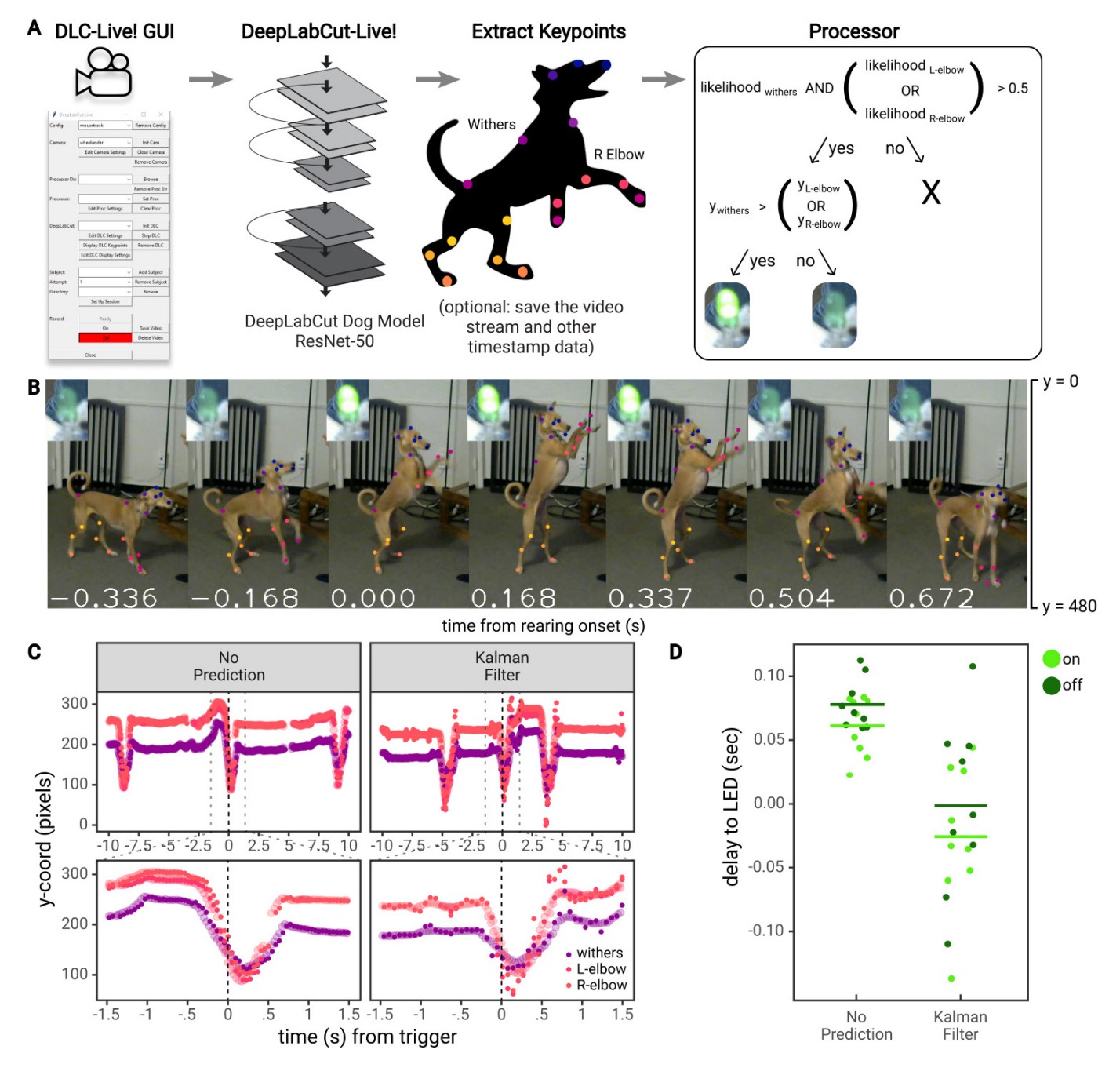

**Figure 6.** Real-time feedback based on posture. (**A**) Diagram of the workflow for the dog feedback experiment. Image acquisition and pose estimation were controlled by the DLC-Live! GUI. A DeepLabCut-live Processor was used to detect 'rearing' postures based on the likelihood and y-coordinate of the withers and elbows and turn an LED on or off accordingly. (**B**) An example jump sequence with LED status, labeled with keypoints measured offline using the DeepLabCut-live benchmarking tool. The images are cropped for visibility. (**C**) Example trajectories of the withers and elbows, locked to one jump sequence. Larger, transparent points represent the true trajectories – trajectories measured offline, from each image in the video. The smaller opaque points represent trajectories measured in real-time, in which the time of each point reflects the delay from image acquisition to pose estimation, with and without the Kalman filter forward prediction. Without forward prediction, estimated trajectories are somewhat delayed from the true trajectories. With the Kalman filter forward prediction, trajectories are less accurate but less delayed when keypoints exhibit rapid changes in position, such as during a rearing movement. (**D**) The delay from the time the dog first exhibited a rearing posture (from postures measured offline) to the time the LED was turned on or off. Each point represents a single instance of the detection of a transition to a rearing posture or out of a rearing posture.

The online version of this article includes the following figure supplement(s) for figure 6:

*Figure 6 continued on next page*

*Figure 6 continued*

**Figure supplement 1.** The Kalman filter predictor reduces errors when predicting up to three frames into the future in a mouse reaching task.
**Figure supplement 2.** The Kalman filter predictor reduces errors when predicting up to three frames into the future in a mouse open-field task.

used this information to predict the position of each keypoint at the current point in time (i.e., not the time at which the image was acquired, but after time had been taken to process the image). For example, if the image was acquired 60 ms ago, the Kalman filter 'Processor' predicted keypoints 60 ms into the future.

Thus, in another set of experiments, we recorded as the dog performed a series of 'rearing' movements, this time using a `Processor` that first forward-predicted the dog's pose, then detected 'rearing' movements and controlled an LED accordingly. Using the `Processor` with a Kalman filter reduced inference speed compared to the `Processor` without the Kalman filter, due to the time required to compute Kalman filter updates (15.616 ± 0.332 frames per second). The image to pose latency was 81 ± 10 ms (N = 1187 frames), and image to LED latency of 82 ± 11 ms (N = 9 'rearing' movements). However, compared to the ideal times to turn the LED on or off calculated from pose estimation performed on all available images, the Kalman filter 'Processor' achieved a delay of −13 ± 61 ms. In this case, the Kalman filter turned the LED on or off 13 ms *prior* to the point at which the first rearing posture was detected (*Figure 6D*). These results indicate the potential to perform zero or negative latency feedback based on posture.

To better understand the utility and limitations of the Kalman filter predictor, we tested the Kalman filter `Processor` offline on two additional datasets: a mouse reaching dataset that exhibited movements on a similar timescale to dog 'rearing' movements (on the order of a few hundred milliseconds), but that was recorded with a higher frame rate (150 FPS), and a mouse open-field dataset that was recorded at the same frame rate as the dog 'rearing' experiment (30 FPS), but exhibited slower, smoother movements. To simplify these analyses, we focused on a single point in both datasets: the 'back of the hand' coordinate involved in the reaching movement, and the 'snout' in the open-field dataset. We examined the accuracy of the Kalman filter when predicting 1–7 frames into the future ( 7–47 ms for the mouse reaching or 33–233 ms for the open field) into the future compared to the accuracy of using the DLC pose that was obtained after a delay of 1–7 frames.

For the mouse reaching dataset, similar to the dog 'rearing' experiment, the Kalman filter predictions eliminated the lag that is present in the delayed pose, but the Kalman filter became more noisy and less accurate as it predicted the position of the back of the hand further into the future (*Figure 6—figure supplement 1*). To quantitatively examine the tracking errors when using the DLC estimated pose delayed by 1–7 frames vs. the Kalman filter predicted pose, we calculated the error as the euclidean distance between the delayed pose or Kalman filter predicted pose and the true DLC estimated pose for each frame throughout an entire experimental session (n = 153,279 frames). We then compared the cumulative distribution function (CDF) of errors (*Figure 6—figure supplement 1*). The CDF indicates the percentage of frames that have errors less than or equal to a particular error value, where a greater percentage of frames at a given error level indicates better performance (or more frames with errors that are smaller than this value). When predicting 4 frames/ 27.6 ms or further into the future, the delayed pose distribution had a greater percentage of frames with smaller errors at all error values, indicating that the Kalman filter produced larger errors than using the delayed pose when predicting 4 or more frames into the future. However, when predicting 1 frame/6.7 ms, 2 frames/13.3 ms, or 3 frames/20 ms into the future, the Kalman filter predicted back of the hand coordinate had more frames with errors smaller than 0.18, 0.85, and 2.94 pixels, respectively. This finding indicates that the Kalman filter predictor reduced the number of frames with larger errors, improving tracking performance when predicting up to 20 ms into the future.

The same analysis of the cumulative distribution of errors was performed on the open field dataset (n = 2330 frames), with similar results (*Figure 6—figure supplement 2*). When predicting 4 frames/133 ms or more into the future, the delayed pose distribution had a greater percentage of frames with smaller errors at nearly all error values. But when predicting 2 frames/67 ms or 3 frames/ 100 ms into the future, the Kalman filter predicted snout coordinate had more frames with errors smaller than 28.61 and 18.34 pixels, respectively, and when predicting 1 frame/33 ms into the future, the Kalman filter predictions had a greater percentage of frames with smaller errors across the entire

distribution. These data indicate that the Kalman filter is effective at predicting further into the future on tracking problems that involve slower, more gradual movements.

## Discussion

Providing event-triggered feedback in real-time is one of the strongest tools in neuroscience. From closed-loop optogenetic feedback to behaviorally-triggered virtual reality, some of the largest insights from systems neuroscience have come through causally testing the relationship of behavior to the brain (*Kim et al., 2017*; *Chettih and Harvey, 2019*; *Jazayeri and Afraz, 2017*). Although tools for probing both the brain and for measuring behavior have become more advanced, there is still the need for such tools to be able to seamlessly interact. Here, we aimed to provide a system that can provide real-time feedback based on advances in deep learning-based pose estimation. We provide new computational tools to do so with high speeds and low latency, as well as a a full benchmarking test suite (and related website: https://deeplabcut.github.io/DLC-inferencespeed-benchmark/), which we hope enables ever more sophisticated experimental science.

### Related work

DeepLabCut and related animal pose estimation tools reviewed in *Mathis and Mathis, 2020* have become available starting in early 2018, and two groups have built tools around real-time applications with DeepLabCut. However, the reported speed and latencies are slower than what we were able to achieve here: *Forys et al., 2020* achieved latencies of 30 ms using top-end GPUs, and this delay increases if the frame acquisition rate is increased beyond 100 frames per second. *Schweihoff et al., 2019* also achieve latencies of 30 ms from frame acquisition to detecting a behavior of interest (round-trip frame to LED equivalent was not reported). We report a 2–3x reduction in latency (11 ms/16 ms from frame to LED in the 'Optimize Latency'/'Optimize Rate' mode of DLG) on a system that uses a less powerful GPU (Windows/GeForce GTX 1080) compared to these studies, and equivalent performance (29 ms/35 ms from frame to LED in the 'Optimize Latency'/'Optimize Rate' mode of DLG) on a conventional laptop (MacBook Pro with Intel Core-i7 CPU). Although we believe such tools can use the advances presented in this work to achieve higher frame rates and lower latencies, our new real-time approach provides an improvement in portability, speed, and latency.

Animal pose estimation toolboxes, like DeepLabCut, have all benefited from advances in human pose estimation research. Although the goals do diverge (reviewed in *Mathis and Mathis, 2020*) in terms of required speed, the ability to create tailored networks, and accuracy requirements, competitions on human pose estimation benchmarks such as PoseTrack (*Andriluka et al., 2018*) and COCO *Lin et al., 2014* have advanced computer vision. Several human pose estimation systems have real-time options: OpenPose (*Cao et al., 2017*) has a real-time hand/face pose tracker available, and Pif-Paf (*Kreiss et al., 2019*) reaches about 10 Hz on COCO (depending on the backbone; *Lin et al., 2014*). On the challenging multi-human PoseTrack benchmark (*Andriluka et al., 2018*), LightTrack (*Ning et al., 2020*) reaches less than 1 Hz. However, recent work achieves 3D multi-human pose estimation at remarkable frame rates (*Chen et al., 2020*), in particular they report an astonishing 154 FPS for 12 cameras with four people in the frame. State of the art face detection frameworks, based on optimized architectures such as BlazeFace can achieve remarkable speeds of >500 FPS on GPUs of cell phones (*Bazarevsky et al., 2019*). The novel (currently unpublished) multi-animal version of DeepLabCut can also be used for feedback, and depending on the situation, tens of FPS for real-time applications should be possible. Inference speed can also be improved by various techniques such as network pruning, layer decomposition, weight discretization or feed-forward efficient convolutions (*Zhang et al., 2019*). Plus, the ability to forward predict postures, as we show here, can be used to compensate for hardware delays.

### Scalability, affordability, and integration into existing pipelines

If neuroscience's embrace of studying the brain in its natural context of complex, contingent, and open-ended behavior (*Krakauer et al., 2017*; *Mathis and Mathis, 2020*; *Datta et al., 2019*) is smashing the champagne on a long-delayed voyage, the technical complexity of the experiments is the grim spectre of the sea. Markerless tracking has already enabled a qualitatively new class of data-dense behavioral experiments, but the heroism required to simultaneously record natural

behavior from 62 cameras (*Bala et al., 2020*) or electrophysiology from 65,000 electrodes (*Sahasrabuddhe et al., 2020*), or integrate dozens of heterogeneous custom-built components (*Findley et al., 2020*) hints that a central challenge facing neuroscience is *scale*.

Hardware-intensive experiments typically come at a significant cost, even if the pose estimation tools are 'free' (developed in laboratories at a non-significant expense, but provided open source). Current commercial systems are expensive–up to $10,000–and they have limited functionality; these systems track the location of animals but not postures or movements of experimenter-defined points of the animal, and few to none offer any advanced deep learning-based solutions. Thus, being able to track posture with state-of-the-art computer vision at scale is a highly attractive goal.

`DeepLabCut-Live!` experiments are a many-to-many computing problem: many cameras to many GPUs (and coordinated with many other hardware components). The Autopilot experiment we described is the simplest 2-computer case of distributed applications of `DeepLabCut-Live!`, but Autopilot provides a framework for its use with arbitrary numbers of cameras and GPUs in parallel. Autopilot is not prescriptive about hardware configuration, so for example if lower latencies were needed users could capture frames on the same computer that processes them, use more expensive GPUs, or use the forward-prediction mode. Along with the rest of its hardware, experimental design, and data management infrastructure, integration of DeepLabCut in Autopilot makes complex experiments scalable and affordable.

Thus, here we presented options that span ultra-high performance (at GPU cost) to usable, affordable solutions that will work very well for most all applications (i.e. up to 90 FPS with zero to no latency if using our forward-prediction mode). Indeed, the Jetson experiments that we performed used simple hardware (inexpensive webcam, simple LED circuit) and either the open source DLC-Live! GUI or AutoPilot.

In addition to integration with Autopilot, we introduce integration of real-time DeepLabCut into Bonsai, a popular framework that is already integrated into many popular neuroscience tools such as OpenEphys (*Siegle et al., 2017*), BonVision (*Lopes et al., 2020*), and BpodDeveloped by Sanworks: https://www.sanworks.io/index.php. The merger of DeepLabCut and Bonsai could therefore allow for real-time posture tracking with sophisticated neural feedback with hardware such as NeuroPixels, Miniscopes, and beyond. For example, Bonsai and the newly released BonVision toolkit (*Lopes et al., 2020*) are tools for providing real-time virtual reality (VR) feedback to animals. Here, we tested the capacity for a single GPU laptop system to run Bonsai-DLC with another computational load akin to what is needed for VR, making this an accessible tool for systems neuroscientists wanting to drive stimuli based on potentially sophisticated postures or movements. Furthermore, in our real-time dog-feedback utilizing the forward-prediction mode we utilized both posture and kinematics (velocity) to be able to achieve sub-zero latency.

## Sharing DLC models

With this paper we also introduce three new features within the core DeepLabCut ecosystem. One, the ability to easily export trained models without the need to share project folders (as previously); two, the ability to load these models into other frameworks aside from DLC-specific tools; and three, we modified the code-base to allow for frozen-networks. These three features are not only useful for real-time applications, but if users want to share models more globally (as we are doing with the DeepLabCut Model Zoo Project *Mathis et al., 2020b*), or have a easy-install lightweight DeepLab-Cut package on dedicated machines for running inference, this is an attractive option. For example, the protocol buffer files are system and framework agnostic: they are easy to load into TensorFlow (*Abadi et al., 2016*) wrappers based on C++, Python, etc. This is exactly the path we pursued for Bonsai's plugin via a C#-TensorFlow wrapperTensorFlowSharp: (https://github.com/migueldeicaza/TensorFlowSharp). Moreover, this package can be utilized even in offline modes where batch processing is desirable for very large speed gains (*Mathis and Warren, 2018*; *Mathis et al., 2020a*).

## Benchmarking `DeepLabCut-Live!` GUI performance

Importantly, the DLG benchmarking data described above was collected by loading images from a pre-recorded video at the same rate that these images were originally acquired, effectively simulating the process of streaming video from a camera in real-time. This method was chosen for a few reasons. First, using a pre-recorded video standardized the benchmarking procedure across different

platforms – it would have been extremely difficult to record nearly identical videos across different machines. Second, reading images from the video in the same exact manner that a video would be recorded from a physical camera in real-time exposed the same delays in DLG as we would observe when recording from a camera in real-time. The delays that we report all occur after images are acquired. Thus, if the desired frame rate is achieved, these benchmarking results are exactly equivalent to the delays that would be observed when recording from a physical camera. The possible frame rates that can be achieved by different cameras are documented by camera manufacturers and the software used to record video from cameras relies on libraries provided by camera manufacturers (e.g. The Imaging Source, The Imaging Source Libraries on GitHub: https://github.com/TheImagingSource) or well-documented open-source projects (e.g. OpenCV, https://github.com/opencv/opencv and Aravis, https://github.com/AravisProject/aravis; *Bradski, 2000*). Additional delays caused by recording from a physical camera are therefore specific to each individual type of camera. Finally, by making the videos used for benchmarking available, DLG users can replicate our benchmarking results, which will help in diagnosing the cause of performance issues.

## Choosing to optimize pose estimation rate vs. latency in the `DeepLabCut-Live!` GUI

When using DLG, if the rate of pose estimation is faster than the rate of image acquisition, pose estimation will run in synchrony with image acquisition (in parallel processes). However, if the pose estimation rate is slower than image acquisition, users have the choice of one of two modes: 'Optimize Rate' or 'Optimize Latency.' In the 'Optimize Rate' mode, pose estimation and image acquisition are run asynchronously, such that pose estimation is run continuously, but pose estimation for a given image does not necessarily start at the time the image was acquired. Thus, the delay from image acquisition to pose estimation is the delay from image acquisition until pose estimation begins plus the time it takes DeepLabCut to estimate the pose. In the 'Optimize Latency' mode, after the pose estimation process finishes estimating the pose on one frame, it will wait for the next frame to be acquired to get back in sync with the image acquisition process. This minimizes the latency from image acquisition to pose estimation, but results in a slower rate of pose estimation, and over course of an experiment, fewer estimated poses. Because the 'Optimize Latency' mode results in fewer estimated poses, we recommend using the 'Optimize Rate' mode for most applications. However, the 'Optimize Latency' mode may be particularly useful for applications in which it is critical to minimize delays to the greatest extent possible and it is not critical to sometimes miss the behavior of interest. One example in which a user may consider the 'Optimize Latency' mode could be to stimulate neural activity upon detecting the tongue exiting the mouth on a subset of trials in a licking task. In this example, the 'Optimize Latency' mode will provide a lower latency from when the image that detects the tongue was acquired to stimulating neural activity. However, the slower pose estimation rate will make it more likely that, on a given trial, the first image in which the tongue is present will not be analyzed. Thus, the 'Optimize Latency' mode will provide the shortest delays on trials in which this image is analyzed, and users can choose to not stimulate on trials in which the tongue is too far outside of the mouth when it is first detected by DeepLabCut.

## Feedback on posture and beyond

To demonstrate the feedback capabilities of `DeepLabCut-Live!` we performed a set of experiments where an LED was triggered based on the confidence of the DeepLabCut network and the posture of the animal (here a dog, but as is DeepLabCut, this package is animal and object agnostic). We also provide a forward-prediction mode that utilizes temporal information via kinematics to predict future postural states. In this demonstration, we used a Kalman filter to obtain filtered estimates of the position, velocity and acceleration at each point in time, and then predicted the future pose via quadratic approximation. We chose this approach for a few reasons: (i) it requires no prior training and (ii) with simple modifications to 2–3 parameters, it can be tailored to suit a wide variety of applications. Since this approach relies on a quadratic approximation, it can be successfully applied to any application for which it is possible to obtain accurate measurements of the position, velocity, and acceleration using a Kalman filter. The performance of the Kalman filter predictor will critically depend on how far into the future one wishes to predict and how quickly the velocity and acceleration of the keypoints change. If there are substantial changes in the velocity or acceleration

of the keypoint within the time frame of the forward prediction, or if a keypoint is not present in most images but suddenly appears (e.g. detecting the appearance of the tongue during a lick sequence), the forward prediction will be inaccurate. This was evident in the dog feedback experiment where we were predicting $\approx 80$ ms into the future during a rapid movement. Using a mouse reaching dataset with movements on a similar timescale, but with video recorded at a much higher rate (150 FPS for mouse reaching vs. 30 FPS for dog), and a mouse open field dataset with video recorded at the same rate as the dog 'rearing' experiment, but with slower movements, we demonstrated that the Kalman filter works extremely well in predicting the future pose at a shorter timescale, and demonstrated the way in which it's predictions become inaccurate as it predicts the pose further into the future. However, despite the inaccurate predictions when the time to predict is longer than the timescale at which velocity and acceleration are changing, the Kalman filter still provides great promise to improve the time to detect the onset of rapid movements.

Additionally, we would like to emphasize that the Kalman filter is only one possible approach to predict the future pose. In addition to demonstrating its utility for forward prediction, the source code for the Kalman filter `Processor` provides a blueprint for implementing different methods of forward prediction using the DeepLabCut-Live! framework tailored for the specific tracking problem. For instance, one can imagine applications to rhythmic movements, where one predicts future behavior from many past cycles. Other time series models such as LSTMs, or neural networks can also be integrated in the predictor class. Furthermore, the simple comparison of the position of 2–3 keypoints is only one possible strategy for detecting the time to trigger peripheral devices. For example, one can build `Processor` objects to trigger on joint angles, or more abstract targets such as being in a particular high dimensional state space. We believe the flexibility of this feedback tool, plus the ability to record long-time scale videos for 'standard' DeepLabCut analysis makes this broadly applicable to many applications.

## Conclusions

We report the development of a new light-weight Python pose estimation package based on DeepLabCut, which can be integrated with behavioral control systems (such as Bonsai and AutoPilot) or used within a new DLC-Live! GUI. This toolkit allows users to do real-time, low-latency tracking of animals (or objects) on high-performance GPU cards or on low cost, affordable and scalable systems. We envision this being useful for precise behavioral feedback in a myriad of paradigms.

# Materials and methods

Alongside this publication we developed several software packages that are available on GitHub. Links are listed in n *Table 4* and *Table 5* and details provided throughout the paper.

## Animals

All mouse work were carried out under the permission of the IACUC at Harvard University (#17-07-309). Dog videos and feedback was exempt from IACUC approval (with conformation from IACUC).

Mice were surgically implanted with a headplate as in *Mathis et al., 2017*. In brief, using aseptic technique mice were anesthetized to the surgical plane, a small incision in the skin was made, the skull was cleaned and dried and a titanium headplate was placed with Metabond. Mice were allowed 7 days to recover and given burphrenorphine for 48 hr post-operatively. Mice used in the licking

**Table 4.** Software packages presented with this paper.

| Name | URL |
| --- | --- |
| `DeepLabCut-Live!` SDK | GitHub Link |
| Benchmarking Submission | GitHub Link |
| Benchmarking Results | Website Link |
| DLC-Live! GUI | GitHub Link |
| Bonsai - DLC Plugin | GitHub Link |
| AutoPilot - DLC | GitHub Link |

**Table 5.** Relevant DLC updates.

| Feature | DLC version | Pub. link |
| --- | --- | --- |
| DLC-ResNets | 1, 2.0+ | *Mathis et al., 2018b*; *Nath et al., 2019* |
| DLC-MobileNetV2s | 2.1+ | *Mathis et al., 2020a* |
| Model Export Fxn | 2.1.8+ | this paper |
| DeepLabCut-live | new package | this paper |

task were trained to lick at fixed intervals (details will be published elsewhere). Mice used in the reaching task were trained as in *Mathis et al., 2017*.

The dog used in this paper was previously trained to perform rearing actions for positive reinforcement, and therefore no direct behavioral manipulation was done for this study.

## DeepLabCut

The mouse DeepLabCut model was trained according to the protocol in *Nath et al., 2019*; *Mathis et al., 2020a*. Briefly, the DeepLabCut toolbox (version 2.1.6.4) was used to (i) extract frames from selected videos, (ii) manually annotate keypoints on selected frames, (iii) create a training dataset to train the convolutional neural network, (iv) train the neural network, (v) evaluate the performance of the network, and (vi) refine the network. This network was trained on a total of 120 labeled frames.

The dog model was initially created based on the 'full_dog' model available from the DeepLabCut Model Zoo (ResNet-50, with ADAM optimization and imgaug augmentation *Jung et al., 2020*; currently unpublished, more details will be provided elsewhere). Prior to running the DLG feedback experiments, initial training videos were taken, frames from these videos were extracted and labeled, and the model was retrained using imgaug with the built in scaling set to 0.1–0.5 to optimize network accuracy on smaller images. This network was re-trained with 157 labeled frames.

After training, DeepLabCut models were exported to a protocol buffer format (.pb) using the new export model feature in the main DeepLabCut package (2.1.8). This can be performed using the command line:

```
dlc model-export /path/to/config.yaml
    or in python:
import deeplabcut as dlc
dlc .export_model("/path/to/config.yaml')
```

## DeepLabCut-Live! package

The DeepLabCut-Live code was written in Python 3 (http://www.python.org), and distributed as open source code on GitHub and on PyPi. It utilizes TensorFlow (*Abadi et al., 2016*), numpy (*Svd et al., 2011*), scipy (*Virtanen et al., 2020*), OpenCV (*Bradski, 2000*), and others. Please see GitHub for complete, platform-specific installation instructions and description of the package.

The DeepLabCut-live package provides a DLCLive class that facilitates loading DeepLabCut models and performing inference on single images. The DLC-Live class also has built in image pre-processing methods to reduce the size of images for faster inference: image cropping, dynamic image cropping around detected keypoints, and image downsizing. DLC-Live objects can be instantiated in the following manner:

```
from dlclive import DLCLive
my_live_object = DLCLive("/path/to/exported/model/directory')

# base instantiation
my_live_object = DLCLive("/path/to/exported/model/directory')

# use only the first 200 pixels in both width and height dimensions of image
```

```
my_live_object = DLCLive("/path/to/exported/model/directory',
                cropping=[0, 200, 0, 200])

# dynamically crop image around detected keypoints, with 20 pixel buffer
my_live_object = DLCLive("/path/to/exported/model/directory',
                dynamic=(True, 0.5, 20))

# resize height and width of image to 1/2 its original size
my_live_object = DLCLive("/path/to/exported/model/directory',
            resize  = 0.5)
```

Inference speed tests were run using a benchmarking tool built into the DeepLabCut-Live package. Different image sizes were tested by adjusting the *pixels* parameter, which specifies the total number of pixels in the image while maintaining the aspect ratio of the full-size image. For all options of the benchmarking tool, please see GitHub or function documentation. Briefly, this tool can be used from the command line:

```
dlc—live—benchmark /path/to/model/directory/ path/to/video/file -o /path/to/
output/directory
```

or from python:

```
from dlclive import benchmark_videos
benchmark_videos ("/path/to/model/directory",
                        "/path/to/video",
            output="/path/to/output/directory")
```

When using dynamic tracking within the `DeepLabCut-Live!` package, if the DeepLabCut reported likelihood of a keypoint is less than the user-defined likelihood threshold (i.e. it is likely that the keypoint was 'lost' from the image), the bounding box around the following image may not include that keypoint. Thus, for the dynamic cropping accuracy analysis, we only analyzed the accuracy of tracking for keypoints that had a likelihood greater than 0.5 on the previous image. This resulted in the exclusion of 2.9%, 7.6%, and 27.6% of all individual keypoints across all images, respective to the bound box size (50, 25, 10).

## DLC-Live! GUI software

The DLC-Live! GUI (DLG) code was also written in Python 3 (http://www.python.org), and distributed as open source code on GitHub and on PyPi. DLG utilizes Tkinter for the graphical user interface. Please see GitHub for complete installation instructions and a detailed description of the package.

DLG currently supports a wide variety of cameras across platforms. On Windows, DLG supports The Imaging Source USB cameras and OpenCV compatible webcams. On MacOS, DLG supports OpenCV webcams, PlayStation Eye cameras (https://github.com/bensondaled/pseyepy) and USB3 Vision and GigE Vision cameras (https://github.com/AravisProject/aravis). On Linux, DLG supports any device compatible with Video4Linux drivers using OpenCV, and USB3 Vision and GigE Vision devices using the Aravis Project.

DLG uses the multiprocess package (*McKerns et al., 2012*) to run image acquisition, writing images to disk and pose estimation in separate processes from the main user-interface. Running these processes in parallel enables users to record higher frame rate videos with minimal sacrifice to pose-estimation speed. However, there are still some delays when running image acquisition and pose-estimation asynchronously: if these processes are run completely independently, the image may not have been acquired immediately before pose-estimation begins. For example, if images are acquired at 100 frames per second, the image will have been acquired with a range of 0–10 ms prior to running pose-estimation on the image. If the pose-estimation process waits for a new image to be acquired then there will be a delay between completing pose-estimation on one image and beginning pose-estimation on the next one. Accordingly, DLG allows users to choose between two modes: (i) the Latency mode, in which the pose-estimation process waits for an a new image to

reduce the latency between image acquisition and pose-estimation and (ii) the Rate mode, in which the pose-estimation process runs independently of image acquisition. In this mode, there will be longer latencies from image acquisition to pose-estimation but the rate of pose-estimation will be faster than in the Latency mode.

To test the performance of DLG, we used a video of a mouse performing a task that required licking to receive reward in the form of a drop of water. Video was collected at 100 frames per second using a The Imaging Source USB3 camera (model number: DMK 37AUX287) and Camera Control software (*Kane and Mathis, 2019*).

We tested the performance of DLG under four conditions– on full-size images (352 × 274 pixels) and downsized images (176 × 137 pixels), both image sizes in Latency mode and Rate mode. All four conditions were tested on four different computers (see *Table 2* for specifications). The mouse licking video used for this test was different from the mouse licking video used for the inference speed benchmarking of the `DeepLabCut-Live!` package (a video with larger images was used in the `DeepLabCut-Live!` benchmark). Across all DLG benchmarking experiments, a single outlier frame-to-LED data point was removed from the Windows OS, GeForce 1080 GPU, 352 × 274 pixel size configuration. This one point had a time frame-to-LED delay of 3 s, nearly 10 times that of the next longest delay under any configuration, likely due to misalignment of the infrared LED and photodetector.

## Inference speed, size vs. accuracy and dynamic cropping

For the analyses regarding the size and accuracy dependency (*Figure 2—figure supplement 2* and *Figure 2—figure supplement 3*), we used the 640 × 480 pixel images available at Zenodo (*Mathis et al., 2018a*) from *Mathis et al., 2018b*.

## Postural feedback experiment

Prior to conducting the dog feedback experiment, the dog was extensively trained to 'rear' upon the verbal command 'jump' and the visual cue of a human's arm raised at shoulder height. 'Rearing' was reinforced by manually providing treats for successful execution. Prior to recording videos for tracking and feedback, the dog routinely participated in daily training sessions with her owners. There was no change to the dog's routine when implementing sessions in which feedback was provided.

To conduct the feedback experiment, we used the DLG software on a Jetson Xavier developer kit. Pose estimation was performed using an exported DeepLabCut model (details regarding training provided above). Feedback was provided using a custom `DeepLabCut-Live! Processor` that detected 'rearing' movements and controlled an LED via serial communication to a Teensy microcontroller.

The dog was considered to be in a 'rearing' posture if (a) the likelihood of the withers and at least one elbow was greater than 0.5 and (b) the vertical position of at least one elbow, whose likelihood was greater than 0.5, was above the vertical position of the withers (i.e., $y_{\mathrm{elbow}} < y_{\mathrm{withers}}$, with the top of the image as $y = 0$ and bottom of image as $y = \mathrm{image\ height}$). For each pose, the `Processor` determined whether the dog was in a 'rearing' posture, queried the current status of the LED from the Teensy microcontroller, and if the current status did not match the dog's posture (i.e. if the LED was off and the dog was in a 'rearing' posture), sent a command to the Teensy to turn the LED on or off.

In this experiment, we recorded the time at which images were accessed by DLG, the time at which poses were obtained by DLG after processing, and the times that the LED was turned on or off by the `Processor`. We calculated the pose estimation rate as the inverse of the delay from obtaining one pose to the next, the latency from image acquisition to obtaining the pose from that image, and, for poses in which the `Processor` turned the LED on or off, the latency from image acquisition to sending the command to turn the LED on or off.

As not all images will be run through pose estimation using DLG, to assess the delay from behavior to feedback, we performed offline analyses to determine the ideal time to turn the LED on or off given all the acquired images. Using the `DeepLabCut-Live!` benchmarking tool, we obtained the pose for all frames in the acquired videos by setting the `save_poses` flag from the command line:

```
dlc -live-benchmark /path/to/model/directory /path/to/video/file –save-poses -
n 0
```

This command can also be run from python:

```
from dlclive import benchmark_videos
benchmark_videos ("/path/to/model/directory",
                      "/path/to/video/file",
                      n_frames = 0,
                      save_poses = True)
```

We then ran this full sequence of poses through the 'rearing' detection `Processor`, and compared these times – for each 'rearing' movement, the time at which the first frame that showed a transition to a 'rearing' posture and out of a 'rearing' posture was acquired – with the times that the LED was turned on or off during real-time feedback.

The videos analyzed in this experiment were different from the dog videos used in the `DeepLab-Cut-Live!` benchmarking experiment. For that experiment, a video with longer duration and different aspect ratio was used.

## Forward prediction using a Kalman filter

To implement the forward-predicting Kalman filter, we used a `Processor` object that first used a Kalman filter to estimate the position, velocity, and acceleration of each keypoint; then used the position, velocity, and acceleration to predict the position of the limb into the future. The Kalman filter was defined by the state vector *X*, consisting of the x and y position, x and y velocity, and x and y acceleration of each keypoint; the forward transition matrix *F*; and the measurement matrix *H*. An example of the full state vector with *n* keypoints is:

$$X = [x_1, ..., x_n, y_1, ..., y_n, ..., \dot{x}_1, ..., \dot{x}_n, \dot{y}_1, ..., \dot{y}_n, \ddot{x}_1, ..., \ddot{x}_n, \ddot{y}_1, ..., \ddot{y}_n]^T$$

For simplicity, we will consider a Kalman filter for a DeepLabCut network with one keypoint:

$$X = [x_1, y_1, \dot{x}_1, \dot{y}_1, \ddot{x}_1, \ddot{y}_1]^T$$

$$F = \begin{bmatrix} 1 & 0 & dt & 0 & \frac{dt^2}{2} & 0 \\ 0 & 1 & 0 & dt & 0 & \frac{dt^2}{2} \\ 0 & 0 & 1 & 0 & dt & 0 \\ 0 & 0 & 0 & 1 & 0 & dt \\ 0 & 0 & 0 & 0 & 1 & 0 \\ 0 & 0 & 0 & 0 & 0 & 1 \end{bmatrix}$$

$$H = \begin{bmatrix} 1 & 0 & 0 & 0 & 0 & 0 \\ 0 & 1 & 0 & 0 & 0 & 0 \\ 0 & 0 & 1 & 0 & 0 & 0 \\ 0 & 0 & 0 & 1 & 0 & 0 \\ 0 & 0 & 0 & 0 & 1 & 0 \\ 0 & 0 & 0 & 0 & 0 & 1 \end{bmatrix}$$

Importantly, performance of the Kalman filter depended on three user-defined scalar variance parameters: the initial variance in state estimates $\sigma_{init}^2$, process noise *Q*, and measurement noise *R*. The initial covariance matrix in state estimates was defined as $P = \sigma_{init}^2 I$.

At each time step, we calculated the estimated pose $X_p$ and estimated covariance $P_p$ based on the previous state:

$$X_p = FX$$

$$P_p = FPF^T + Q$$

We noticed that predictions tended to be inaccurate when the predicted velocity and acceleration were very high. To encourage lower estimates of velocity and acceleration in the estimated state vector $X_p$, we introduced a priori assumption that, at any given timestep, velocity and acceleration will have a zero mean with a user-defined variance $B$. We incorporated this prior assumption using Bayes rule, such that the velocity and acceleration components of $X_p$ were the weighted mean of 0 and $X_p$. The weight depended on the ratio of the variance in the estimate of $X_p$ (the diagonals of the covariance matrix $P$, referred to as $P_d$) and the variance of the prior given by $B$:

$$X_p = \frac{1}{P_d^{-1} + B^{-1}} \cdot \frac{X}{P_d}$$

The observation vector $y$ consisted of the observed position, velocity, and acceleration. The observed position was taken as the pose returned by the DeepLabCut network, the observed velocity was calculated as the change in position from the observed position and position components of the latest state vector $X$, and the acceleration was calculated as the change between the observed velocity and the velocity components of the latest state vector $X$. Next, we updated the state vector $X$ and covariance matrix $P$ according to the Kalman gain $K$:

$$K = P_p H^T \bullet (H P_p H^T + R)^{-1}$$

$$X = X_p + K(y - H X_p)$$

$$P = (I - K H^T) P_p$$

Given this estimate of the current position, velocity and acceleration (the state vector $X$), we used the forward transition matrix $F$ to calculate the predicted future state $X_f$. In the dog feedback experiment, the amount of time we predicted into the future depended on the delay for that image ($\text{dt} = \text{prediction time} = \text{current time} - \text{image acquisition time}$):

$$X_f = FX$$

To obtain the future pose, we extracted the position elements from $X_f$, and discarded the velocity and acceleration components. Lastly, the `Processor` checked if the dog was in a 'rearing' posture and controlled the LED accordingly.

Source code for the base Kalman filter Processor and the dog rearing Processor can be found on Github. Additionally, the Kalman filter predicting `Processor` is in the main `DeepLabCut-Live!` package, and can be used as follows:

```
from dlclive .processor import KalmanFilterPredictor
```

## Details of AutoPilot setup

Latencies were measured using software timestamps and confirmed by oscilloscope. Software measurements could be gathered in greater quantity but were reliably longer than the oscilloscope latency measurements by 2.8 ms (median of difference, n = 75, IQR=[2.4–3.4]), thus we use the software measurements noting they are a slightly conservative estimate of functional latency.

Autopilot experiments were performed using the DLC_Latency Task and the Transformer 'Child' (see *Saunders and Wehr, 2019* for terminology).

Separate DLC-MobileNetV2-0.35 models tracking a single point were trained for each capture resolution (128 × 128, 256 × 256, 512 × 416 pixels). Training data was labeled such that the point was touching the LED when it was on, and in the corner of the frame when the LED was off. Frames were processed with a chain of Autopilot `Transform` objects like:

```
from autopilot import transform as t

# create transformation object
tfm =t.image.DLC("/model/path') + \
```

```
        t.selection
        t.logical.Condition(
            minimum = [min_x, min_y],
            maximum = [max_x, max_y]
        )
# process a frame, yielding a bool
# true/false == LED on/off
led_on = tfm.process(frame)
```

where `min_x`, `min_y`, etc. defined a bounding box around the LED.

## Data analysis and visualization

Autopilot data were analyzed with Pandas (1.0.5; *McKinney et al., 2010*) in Python and Tidyverse (1.3.0; *Wickham et al., 2019*) in R. Data were visualized with ggplot2 (3.3.0; *Wickham, 2016*) and ggridges (0.5.2; *Wilke, 2020*).

## Acknowledgements

We thank Sébastien Hausmann for testing the `DLC-Live!` GUI, and to Jessy Lauer for optimization assistance. We thank Jessica Schwab and Izzy Schwane for assistance with the dog feedback experiment. We greatly thank Michael Wehr for support of AutoPilot-DLC, and the Mathis Lab for comments. Funding was provided by the Rowland Institute at Harvard University and the Chan Zuckerberg Initiative DAF, an advised fund of Silicon Valley Community Foundation to AM and MWM; a Harvard Mind, Brain, Behavior Award to GK, MWM; and NSF Graduate Research Fellowship No. 1309047 to JLS. The authors declare no conflicts of interest. Contributions GK, AM, MWM conceptualized the project. GK, JS, GL, AM, MWM designed experiments. GK and JS developed Jetson integration. GK developed and performed experiments for DLC-Live, and the DLC-Live GUI. JS developed and performed experiments for AutoPilot-DLC. GL developed and performed experiments for Bonsai-DLC. AM contributed to Bonsai-DLC and DLC-Live. All authors performed benchmarking tests. GK and MWM original draft, all authors contributed to writing. MWM supervised the project.

## Additional information

### Competing interests

Gonçalo Lopes: Gonçalo Lopes is director at NeuroGEARS Ltd. The other authors declare that no competing interests exist.

### Funding

| Funder | Grant reference number | Author |
| --- | --- | --- |
| Chan Zuckerberg Initiative | EOSS | Alexander Mathis Mackenzie W Mathis |
| National Science Foundation | 1309047 | Jonny L Saunders |
| Rowland Institute at Harvard | | Gary A Kane Alexander Mathis Mackenzie W Mathis |
| Harvard Brain Science Initiative | | Gary A Kane Mackenzie W Mathis |

The funders had no role in study design, data collection and interpretation, or the decision to submit the work for publication.

## Author contributions
Gary A Kane, Conceptualization, Data curation, Software, Formal analysis, Investigation, Visualization, Methodology, Writing - original draft, Writing - review and editing; Gonçalo Lopes, Software, Formal analysis, Investigation, Methodology, Writing - review and editing; Jonny L Saunders, Software, Formal analysis, Investigation, Methodology, Writing - original draft, Writing - review and editing; Alexander Mathis, Conceptualization, Software, Formal analysis, Investigation, Methodology, Writing - review and editing; Mackenzie W Mathis, Conceptualization, Software, Formal analysis, Supervision, Funding acquisition, Visualization, Methodology, Writing - original draft, Project administration, Writing - review and editing

## Author ORCIDs
Gary A Kane (iD) http://orcid.org/0000-0002-7703-5055
Gonçalo Lopes (iD) https://orcid.org/0000-0003-0731-4945
Jonny L Saunders (iD) https://orcid.org/0000-0003-0545-5066
Alexander Mathis (iD) https://orcid.org/0000-0002-3777-2202
Mackenzie W Mathis (iD) https://orcid.org/0000-0001-7368-4456

## Ethics
Animal experimentation: All mouse work was carried out under the permission of the IACUC at Harvard University (#17-07-309). Dog videos and feedback was exempt from IACUC approval (with conformation with IACUC).

## Decision letter and Author response
Decision letter https://doi.org/10.7554/eLife.61909.sa1
Author response https://doi.org/10.7554/eLife.61909.sa2

---

# Additional files
## Supplementary files
• Transparent reporting form

## Data availability
All models, data, test scripts and software is released and made freely available on GitHub: https://github.com/DeepLabCut/DeepLabCut-live (copy archived at https://archive.softwareheritage.org/swh:1:rev:02cd95312ec6673414bdc4ca4c8d9b6c263e7e2f/).

---

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
