## [Decision Letter]

**Acceptance summary:**

This submission introduces a new set of software tools for implementing real-time, marker-less pose tracking. The manuscript describes these tools, presents a series of benchmarks, and demonstrates their use in several experimental settings: including deploying low-latency closed-loop events triggered on an animal's pose. The first key development presented is a new python package – DeepLabCut-Live! – which optimizes pose inference to increase its speed, a key step for real-time application of DLC. The authors then present a new method for exporting trained DLC networks in a language-independent format and demonstrate how these can be used in three different environments to deploy experiments. Importantly, in addition to developing their own GUI, the authors have developed plugins for Bonsai and AutoPilot, two software packages already widely used by the systems neuroscience community to run experiments.

**Decision letter after peer review:**

Thank you for submitting your article "Real-time, low-latency closed-loop feedback using markerless posture tracking" for consideration by *eLife*. Your article has been reviewed by three peer reviewers, including Gordon Berman as the Reviewing Editor and Reviewer #1, and the evaluation has been overseen by Timothy Behrens as the Senior Editor. The following individual involved in review of your submission has agreed to reveal their identity: Tiago Branco (Reviewer #2).

The reviewers have discussed the reviews with one another, were largely positive about the article, and the Reviewing Editor has drafted this decision to help you prepare a revised submission.

This submission introduces a new set of software tools for implementing real-time, marker-less pose tracking. The manuscript describes these tools, presents a series of benchmarks and demonstrates their use in several experimental settings, which include deploying very low-latency closed-loop events triggered on pose detection. The software core is based on DeepLabCut (DLC), previously developed by the senior authors. The first key development presented is a new python package – DeepLabCut-Live! – which optimizes pose inference to increase its speed, a key step for real-time application of DLC. The authors then present a new method for exporting trained DLC networks in a language-independent format and demonstrate how these can be used in three different environments to deploy experiments. Importantly, in addition to developing their own GUI, the authors have developed plugins for Bonsai and AutoPilot, two software packages already widely used by the systems neuroscience community to run experiments.

All three reviewers agreed that this work is exciting, carefully done, and would be of interest to a wide community of researchers. The consensus was the this was a strong submission. There were, however, four points that the reviewers felt could be addressed to increase the scope and the influence of the work (enumerated below).

Essential revisions:

1) The fundamental trade-off in tracking isn't image size vs. speed, but rather accuracy vs. speed. Thus, the reviewers felt that providing a measure of how the real (i.e., not pixel space) accuracy of the tracking was affected by changing the image resolution would be very helpful to researchers wishing to design experiments that utilize this software.

2) The manuscript would also benefit from including additional details about the Kalman filtering approach used here (as well as, potentially, further discussion about how it might be improved in future work). For instance, while the use of Kalman Filters to achieve sub-zero latencies is very exciting, it is unclear how robust this approach is. This applies not only to the parameters of the filter itself, but also on the types of behavior that this approach can work with successfully. Presumably, this requires a high degree of stereotypy and reproducibility of the actions being tracked and the reviewers felt that some discussion on this point would be valuable.

3) A general question that the reviewers had was how the number of key (tracked) points affects the latency. For example, the “light detection task” using AutoPilot uses a single key-point, how would the additional of more key-points affect performance in this particular configuration? More fully understanding this relationship would be very helpful in guiding future experimental design using the system.

4) The DLG values appear to have been benchmarked using an existing video as opposed to a live camera feed. It is conceivable that a live camera feed would experience different kinds of hardware-based bottlenecks that are not present when streaming in a video (e.g., USB3 vs. ethernet vs. wireless). Although this point is partially addressed with the demonstration of real-time feedback based on posture later in the manuscript, a replication of the DLG benchmark with a live stream from a camera at 100 FPS would be helpful to demonstrate frame rates and latency given the hardware bottlenecks introduced by cameras. If this is impossible to do at the moment, however, at minimum, adding a discussion stating that this type of demonstration is currently missing and outlining these potential challenges would be important.

Again, though, the reviewers were very enthusiastic about this Tools and Resources submission, and the above points primarily point towards ways in which the authors could increase the manuscript's impact, rather than fundamental concerns about its validity or overall value to the community.

---

## [Author Response]

Essential revisions:1) The fundamental trade-off in tracking isn't image size vs. speed, but rather accuracy vs. speed. Thus, the reviewers felt that providing a measure of how the real (i.e., not pixel space) accuracy of the tracking was affected by changing the image resolution would be very helpful to researchers wishing to design experiments that utilize this software.

We thank the reviewers for this important point! Indeed, accuracy can be affected when images are downsampled, if this is not accounted for during network training. Accuracy could be affected for two reasons: (i) downsampling the image lowers the resolution significantly, and this loss of information may lead to inaccurate predictions, or (ii) the network may not have been trained on smaller images. Mathis and Warren, 2018, has shown that DeepLabCut is mostly robust to loss of information due to video compression (see Mathis and Warren, 2018, Figure 3). However, when images are compressed to a great degree, accuracy is reduced. Similarly, we expect that DeepLabCut will be robust to downsizing images within reason, but that accuracy will be reduced if images are downsized to a great degree. However, reduced accuracy may be mitigated by training networks specifically on downsized images. If someone knows they want to use the network in a more downsampled setting, they can train with a different range. Within DeepLabCut, this can easily be set in the `pose_cfg.yaml` file by modifying the default parameters:

`scale_jitter_lo: 0.5` and/or `scale_jitter_up: 1.25` before training (see more here: https://github.com/DeepLabCut/DeepLabCut/blob/8115ee24750b573bce4eb987a27010a5b 6cf2557/deeplabcut/pose_cfg.yaml#L44). Note, in this work, we already account for users to downsample to ~0.5 and up to ~1.25 as this is the default for training all DLC networks.

However, to better understand how downsizing images affects tracking accuracy, and to demonstrate this relationship to users, we directly tested accuracy across a range of downsized images on networks that were trained using the default scale parameters (scale = 0.5-1.25), a wider range of image sizes (scale = 0.1-1.25), and a network trained only on smaller images (scale = 0.1-0.5). A new subsection of the Results “Inference speed: size vs. accuracy” has been added (and quoted below), and we added a new supplementary figure, Figure 2—figure supplement 2:

“Although reducing the size of images increases inference speed, it may result in reduced accuracy in tracking if the resolution of the downsized image is too small to make accurate predictions, or if the network is not trained to perform inference on smaller images. […] Overall, this highlights for one example that downsizing might not strongly impact the accuracy and that this might be the easiest way to increase inference speeds”.

Relatedly, we want to highlight a feature that allows the user to still run “full size” frames as they wish, but only run online analysis in a dynamically cropped way. I.e., if you wish to track the pupil, or a hand reaching, for example, you can set up a box around just these smaller areas, and even if the animal moves, DLC will “dynamically crop” to that size and find the pupil or hand. Thus, small frame sizes can be very attractive for many applications – the benefit of full resolution (higher accuracy), with the speed of a smaller area.

We now include a new subsection in the results “Dynamic cropping: increased speed without effects on accuracy”, along with a new supplementary figure, Figure 2—figure supplement 3, on this feature.

2) The manuscript would also benefit from including additional details about the Kalman filtering approach used here (as well as, potentially, further discussion about how it might be improved in future work). For instance, while the use of Kalman Filters to achieve sub-zero latencies is very exciting, it is unclear how robust this approach is. This applies not only to the parameters of the filter itself, but also on the types of behavior that this approach can work with successfully. Presumably, this requires a high degree of stereotypy and reproducibility of the actions being tracked and the reviewers felt that some discussion on this point would be valuable.

We agree that the manuscript could benefit from both additional details about the Kalman filter prediction and a further discussion about its applications and limitations. We added complete mathematical details about the Kalman filter estimation of the position, velocity, and acceleration of each keypoint and how they are used to predict the future position of each keypoint. This can be found in the Materials and methods subsection “Postural feedback experiment”, “Forward prediction using the Kalman Filters”.

Regarding limitations of the Kalman filter, we do want to clarify this point: “Presumably, this requires a high degree of stereotypy and reproducibility of the actions being tracked.” In fact, there is no formal requirement for the behavior to be stereotypical. The forward model is not trained directly on the data beforehand -- the forward prediction is simply a quadratic approximation using the position, velocity and acceleration. We encourage the user to try this on the behavior beforehand, but there is no inherent limitation, provided that the time the user wishes to predict in the future is not significantly greater than the timescale at which velocity and acceleration of the keypoints are changing. We edited our discussion about the postural feedback experiment to include a more detailed discussion regarding the use-cases and limitations of the Kalman filter:

“To demonstrate the capabilities of DeepLabCut-Live! we performed a set of experiments where an LED was triggered based on the confidence of the DeepLabCut network and the posture of the animal (here a dog, but as is DeepLabCut, this package is animal and object agnostic). […] We believe the flexibility of this feedback tool, plus the ability to record long-time scale videos for “standard” DeepLabCut analysis makes this broadly applicable to many applications.“

We also further demonstrate the utility of the Kalman filter on two additional datasets: a mouse reaching dataset with rapid movements on the order of a few hundred milliseconds, and a mouse open-field dataset with slower, more gradual movements.

3) A general question that the reviewers had was how the number of key (tracked) points affects the latency. For example, the “light detection task” using AutoPilot uses a single key-point, how would the additional of more key-points affect performance in this particular configuration? More fully understanding this relationship would be very helpful in guiding future experimental design using the system.

Thank you for raising this concern. To address this point, we directly tested how the number of keypoints affects inference speed using the DeepLabCut-Live! Package, as this would directly affect any downstream packages, like AutoPilot. These results are reported in a new subsection of the Results, “Number of keypoints does not affect inference speed”, and in a supplementary figure, Figure 2—figure supplement 4:

“We also tested whether the number of keypoints in the DeepLabCut network affected inference speeds. […] The number of keypoints in the model had no effect on inference speed for either backbone (Figure 2—figure supplement 4).”

4) The DLG values appear to have been benchmarked using an existing video as opposed to a live camera feed. It is conceivable that a live camera feed would experience different kinds of hardware-based bottlenecks that are not present when streaming in a video (e.g., USB3 vs. ethernet vs. wireless). Although this point is partially addressed with the demonstration of real-time feedback based on posture later in the manuscript, a replication of the DLG benchmark with a live stream from a camera at 100 FPS would be helpful to demonstrate frame rates and latency given the hardware bottlenecks introduced by cameras. If this is impossible to do at the moment, however, at minimum, adding a discussion stating that this type of demonstration is currently missing and outlining these potential challenges would be important.

Thank you for raising this point:  “It is conceivable that a live camera feed would experience different kinds of hardware-based bottlenecks that are not present when streaming in a video (e.g., USB3 vs. ethernet vs. wireless).” We have addressed this concern in a new section of the Discussion, “Benchmarking DeepLabCut-Live! GUI Performance,”:

“Importantly, the DLG benchmarking data described above was collected by loading images from a pre-recorded video at the same rate that these images were acquired, effectively simulating the process of streaming video from a camera in real-time. […] Finally, by making the videos used for benchmarking available, DLG users can replicate our benchmarking results, which will help in diagnosing the cause of performance issues.”